# CLAP: COLLABORATIVE ADAPTATION FOR PATCHWORK LEARNING

**Sen Cui**[*1] **Wuerkaixi Abudukelimu**[*1] **Weishen Pan**[2] **Jian Liang**[3]
**Lei Fang**[4] **Changshui Zhang**[†1] **Fei Wang**[†2]
[1] Institute for Artificial Intelligence, Tsinghua University (THUAI)
Beijing National Research Center for Information Science and Technology (BNRist)
Department of Automation, Tsinghua University, Beijing, P.R.China
[2] Department of Population Health Sciences,
Weill Cornell Medical College, Cornell University, New York, USA
[3] Independent Researcher
[4] DataCanvas Technology Co., Ltd.

## ABSTRACT

In this paper, we investigate a new practical learning scenario, where the data distributed in different sources/clients are typically generated with various modalities. Existing research on learning from multi-source data mostly assume that each client owns the data of all modalities, which may largely limit its practicability. In light of the expensiveness and sparsity of multimodal data, we propose "patchwork learning" to jointly learn from fragmented multimodal data in distributed clients. Considering the concerns on data privacy, patchwork learning aims to impute incomplete multimodal data for diverse downstream tasks without accessing the raw data directly. Local clients could miss different modality combinations. Due to the statistical heterogeneity induced by non-i.i.d. data, the imputation is more challenging since the learned dependencies fail to adapt to the imputation of other clients. In this paper, we provide a novel imputation framework to tackle modality combination heterogeneity and statistical heterogeneity simultaneously, called "collaborative adaptation". In particular, for two observed modality combinations from two clients, we learn the transformations between their maximal intersection and other modalities by proposing a novel ELBO. We improve the worst-performing required transformations through a Pareto min-max optimization framework. In extensive experiments, we demonstrate the superiority of the proposed method compared to existing related methods on benchmark data sets and a real-world clinical data set.

## 1 INTRODUCTION

Multi-modal learning (Ngiam et al., 2011), which refers to the paradigm of learning from the data with multiple modalities, has gained growing interest for its practical significance in facilitating real-world applications (Valindria et al., 2018; Yang et al., 2022). In reality, multimodal data are typically generated from various users/clients with private information. Given the costly and sparse nature of such multimodal data, an interesting problem is how to learn from distributed multimodal data without sacrificing privacy.

One challenge of learning from distributed multimodal data is modality heterogeneity. In many applications, local data on clients are generated with various modalities. More importantly, it is usually hard to require all modalities to exist in each client in reality. One example is collaborative learning in the clinical research network (CRN) involving multiple hospitals (Fleurence et al., 2014), where each hospital can be viewed as a local client. If we want to collaboratively build a mortality prediction model for COVID-19 patients, different hospitals can have various medical records due to

[*] These authors contributed equally to this work.
[†] Corresponding authors

different diagnoses and treatments. Hence, it becomes crucial to enhance scalability by dealing with different combinations of modalities.

Another challenge, which has been studied in federated learning (FL)(McMahan et al., 2017), is statistical heterogeneity (i.e., non-identically distributed data from local clients). As the remote data sources could be gathered from various users in reality, there is research showing that a global model can suffer severe performance degradation when the local data distributions drift dramatically (Deng et al., 2020).

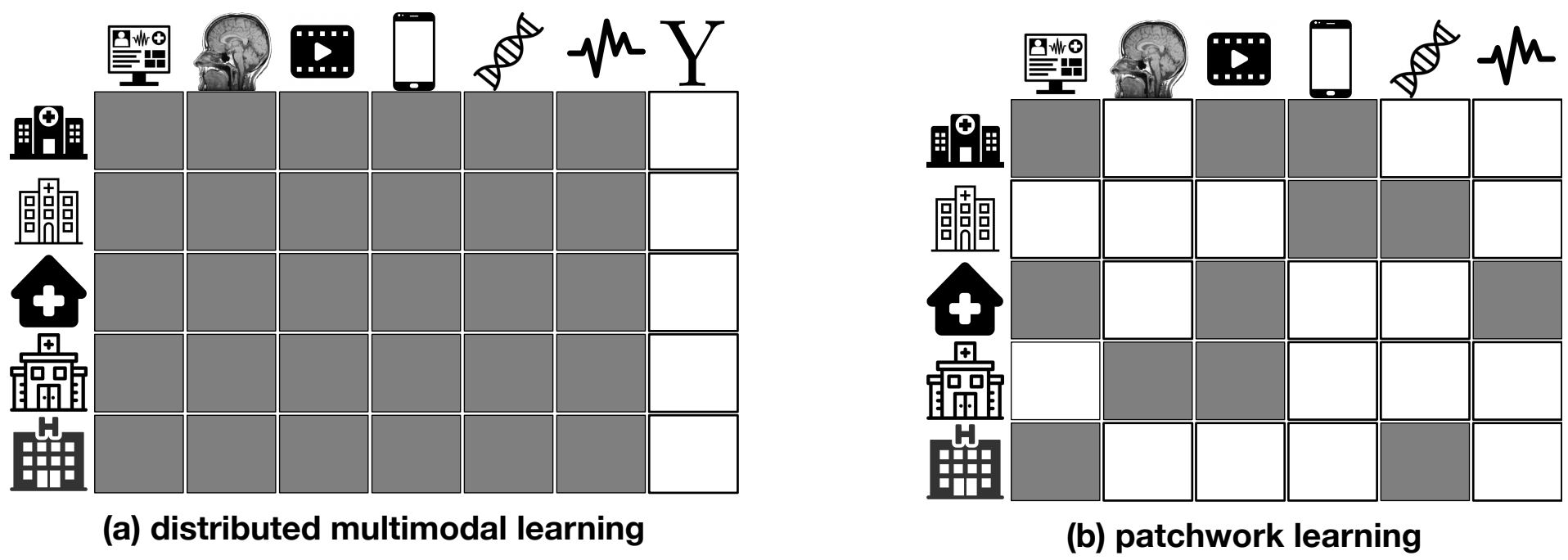

Figure 1: Illustrations of the two learning paradigms. Gray grids denote the observed modalities and white grids are missing modalities. Y is the label to be predicted. (a). distributed multimodal learning; the data in local clients are multimodal and all modalities are observed. (b). patchwork learning; different clients have different multimodal data. The goal is to jointly learn from the "patchwork" to impute all missing modalities without direct access to the raw data.

With the aforementioned considerations, we propose a general multimodal learning problem in a practical setting, called patchwork learning (Rajendran et al., 2023), where data from distributed clients have different modality combinations. Instead of learning a prediction model using complete multimodal data in existing distributed multimodal learning (Zhao et al., 2021; Xiong et al., 2022), where each client has the data of all modalities shown in Figure 1 (a), patchwork learning aims to learn from incomplete data as shown in Figure 1 (b). Each client owns one or multiple modalities. Different clients have different modality combinations, which introduces additional difficulty when collaboratively learning models. Our goal is to impute the unobserved modalities to provide convenience for various downstream tasks without direct access to the raw data.

Modality imputation from observed data becomes even more challenging when it associates with statistical heterogeneity. As the missing modalities of a specific target client are unavailable during training, the imputation process depends on the learned dependencies among modalities from other clients, known as source clients. This may bring bias because of the statistical heterogeneity. To tackle the statistical heterogeneity and modality heterogeneity simultaneously, we propose a novel imputation framework called "collaborative adaptation" (`CLAP`). For statistical heterogeneity, we propose to learn a common modality VAE for each modality for all clients. We mitigate the distribution discrepancy of the generated representations from modality VAEs by balancing the distribution distance of all clients. For different modality combinations in local clients, we propose **client-adaptation VAE** (CA-VAE) for approximating the required dependencies between the maximal intersection of observed modalities in source and target clients, which maximally utilizes the observed modalities for the missing modality imputation.

We empirically evaluate our framework on a series of benchmark data sets, and a real-world electronic health record (EHR) data repository eICU, which includes patient EHR data in ICU from multiple hospitals. The results show our framework outperforms existing relevant methods in the modality imputation tasks and classification downstream tasks. The source codes of our framework are made publicly available at `https://github.com/zaocan666/CLAP`.

## 2 RELATED WORK

Learning from diverse modalities has the potential to improve the performance of machine learning algorithms as they could provide complementary information. For example, the visual appearance and tactile impression of an object could converge on a more invariant abstract characterization (Yildirim, 2014). While there are research aiming to bridge modalities with fully-supervised

complete data (Pandey & Dukkipati, 2017; Ledig et al., 2017), an important problem is that multi-modal data is usually expensive and sparse (Wu & Goodman, 2018). To overcome this obstacle, Wu *et al.* firstly propose MVAE(Wu & Goodman, 2018) to yield generalizable representations to capture the relationship across modalities in a weakly supervised setting, where there is only a subgroup of examples with all modalities present. MVAE flexibly handles missing data in a scalable way for arbitrary multiple views. Recently, several variants of MVAE are proposed to strengthen the generation quality and the coherence between modalities (Shi et al., 2019; Sutter et al., 2020b;a; Hwang et al., 2021; Daunhawer et al., 2021). As existing multimodal representation learning methods were proposed for traditional learning scenario, there is no distribution discrepancies and privacy concerns, which limit their practicability in patchwork learning scenario.

The concept of federated learning (FL) was first proposed by McMahan *et al.* (McMahan et al., 2017). Recently, statistical heterogeneity in FL has drawn much attention in machine learning community and a variety of efforts have been made to tackle it. There are research proposing to calibrate local objectives by regulating the deviation between local models and the global model (Li et al., 2021; Smith et al., 2017). For example, Smith *et al.* aim to model the information-sharing mechanism between local objectives by a multi-task regularization(Smith et al., 2017). Another line of research focus on a better trade-off between local and global learning (Liang et al., 2020; Collins et al., 2021; Khodak et al., 2019; Fallah et al., 2020). For example, some studies assume the heterogeneity comes from label shift and propose to use a multi-head network to model a common feature embedding (Collins et al., 2021). In existing work, it is usually assumed that data from all clients share a common feature space. Besides statistical heterogeneity, a more universal problem is *how to learn when different clients own diverse data modalities*.

Considering the diverse data modalities in real life, there are a few research investigating the task of *distributed multi-modal learning*, i.e., collaboratively learning models on distributed sources containing multimodal data (Xiong et al., 2022; Zhao et al., 2021). In particular, Zhao *et al.* present a multimodal FedAvg algorithm to aggregate the representation extracted by local autoencoders for downstream tasks(Zhao et al., 2021). Xiong *et al.* propose a co-attention mechanism(Xiong et al., 2022) to fuse different modalities under the assumption that the multimodal data are completely observed as shown in Figure 1 (a). In this work, we are interested in addressing a more practical and upstream learning task: how to impute a patchwork when local clients have various incomplete multimodal data.

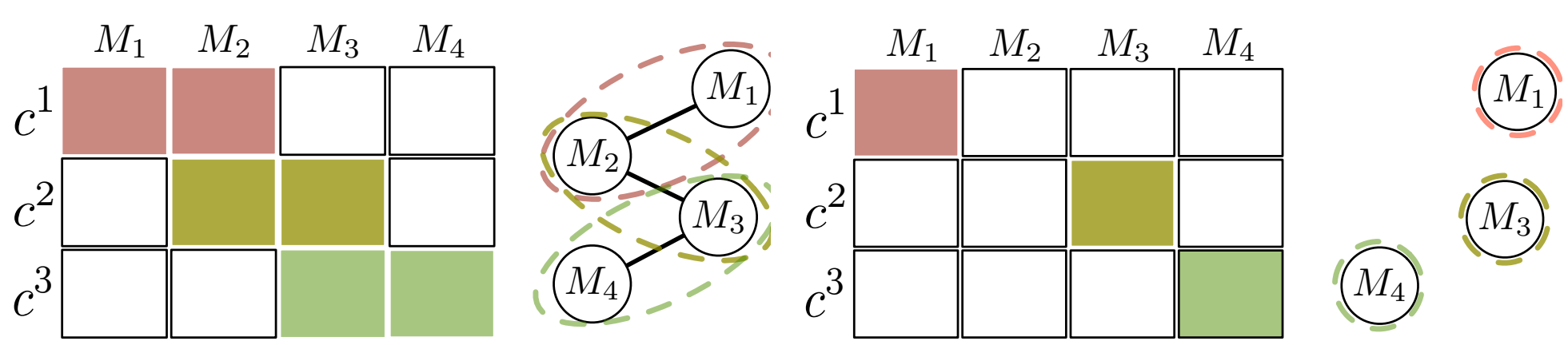

Figure 2: Illustrations of Assumption 2. (a). connected sub-graph; $\{M_1, M_2\}$ is a connected sub-graph, because $c^1$ has both modalities. Similarly, $\{M_2, M_3\}$ ($\{M_3, M_4\}$) is a connected sub-graph because of $c^2$ ($c^3$). (b). no connected sub-graph; $M_1$, $M_3$ and $M_4$ are isolated points.

## 3 Notations and Problem Definition

### 3.1 Notations

Suppose there are $N$ clients with $K$ modalities in a patchwork. The input space $X^i$ is not shared across all clients. Different local clients can have different subsets of all modalities. An example is shown in Figure 1(b), $X^1 = \{M_1, M_3, M_4\}$ but $X^2 = \{M_4, M_5\}$. We use $x^i = \{m_1^i, m_2^i, ..., m_K^i\}$ to denote the sample variable of $X^i$, where $m_j^i$ is the $j$-th modality in the $i$-th client. For example, $x^1 = \{m_1^1, m_3^1, m_4^1\}$ is the sample variable of $X^1$ shown in Figure 1 (b). The output space $Y^i$ denotes the missing modalities. Note that the missing modalities $Y^i$ are not accessible during training.

### 3.2 PATCHWORK LEARNING

Suppose each modality $M_p$ is a node of a graph, and there is an undirected edge from $M_p$ to $M_q$ if there exists a client $c^i$ owning both modalities. An edge between $M_p$ and $M_q$ means that we could model the dependency between them.

To enable the possibility of the imputation of blank grids in a patchwork, we propose the following assumptions.

**Assumption 1.** *For the row in the patchwork, each client owns the data with at least one modality.*

Assumption 1 means that the observed modalities cannot be an empty set for each client.

**Assumption 2.** *There exists a connected sub-graph in the patchwork.*

We give an example shown in Figure 2(a). When there exist sub-graphs in the patchwork, it guarantees the dependency between missing modalities $M_p \in Y^i$ and the observed modalities $X^i$ could be learned from other clients.

A missing modality data $m_j^i$ in the client $c^i$ is imputed relying on the learned dependency between $M_j$ and the observed modalities $X^i$, and the completed imputation of the patchwork means that all missing modalities could be imputed. We have the following proposition.

**Proposition 1.** *(informal, proof in Appendix) Assumption 1 and a connected graph are the necessary and sufficient conditions for the completed imputation of the patchwork.*

Since $Y^i$ is not available during training, the imputation of the target client $c^i$ relies on the dependencies learned from other clients. In general, patchwork learning faces the following challenges that need to be addressed:

- **statistical heterogeneity;** the multimodal data of local clients are typically non-i.i.d. The model may fail to adapt the learned dependencies to target clients;
- **modality combination heterogeneity;** local clients can have various modality combinations. The learned dependencies from $X^i$ is hard to be used for other clients with different combinations $X^j$;
- **modality combination vulnerability;** the learned imputation method could be vulnerable to the modality combinations and the imputation quality significantly varies for two similar combinations;

## 4 METHODOLOGY

### 4.1 PRELIMINARIES: VARIATIONAL AUTOENCODER (VAE), PRODUCT-OF-EXPERTS (POE) AND PARETO MIN-MAX OPTIMIZATION

**VAE.** VAE (Kingma & Welling, 2013) is proposed as a generative model, which assumes the latent variable $z$ is sampled from a prior standard normal distribution $p(z) := N(0, 1)$. VAE consists of an encoder $q_\theta(z|x)$ with the parameters $\theta$, and a decoder $p_\phi(x|z)$ with the parameters $\phi$. During training, VAE maximizes $\log p(x)$ by optimizing the evidence lower bound (ELBO) formulated as

$$\max_{\theta,\phi} \text{ELBO}(x),\ where\ \text{ELBO}(x) := \mathbb{E}_{q_\theta(z|x)} [\log p_\phi(x|z)] - D_{KL} (q_\theta(z|x), p(z)), \quad (1)$$

where $D_{KL} (q_\theta(z|x), p(z))$ denotes the Kullback-Leibler divergence between $q_\theta(z|x)$ and $p(z)$.

**PoE.** Wu *et al.* propose PoE(Wu & Goodman, 2018) for multimodal variational autoencoder (MVAE). MVAE assumes that the multimodal data are conditionally independently generated from a joint latent representation $z$. Suppose there are $K$ modalities $X = \{M_1, M_2, ..., M_K\}$. The generative model has the following factorized form:

$$p(m_1, m_2, ..., m_K, z) = p(z)p(m_1|z)\,p(m_2|z)\,...p(m_K|z). \quad (2)$$

According to Eq.(2), Wu *et al.* propose to employ PoE to aggregate the representations of multiple modalities. For any modality combination $X^s \subset X$, PoE obtains the aggregated representation $q(z|x^s)$ as follows.

$$q(z|x^s) = \frac{\prod_{m_i \in x^s} q(z|m_i)\,p(z)}{\prod_{i=1, m_i \in x^s}^{K} p(z)}. \quad (3)$$

where $q(z|m_i)$ and $p(z)$ are both Gaussian, and the product and quotient distributions are solvable in closed form[1].

**Pareto min-max optimization.** Suppose $\boldsymbol{l}(h) = [l_1(h), l_2(h), \dots l_N(h)]$ represents the loss vector on $N$ learning tasks with hypothesis $h$, $h$ is a Pareto solution if there is no hypothesis $h'$ that dominates $h$: $h' \prec h$, i.e.,

$$\nexists h' \in \mathcal{H}, \text{ s.t. } \forall i : l_i(h') \leq l_i(h) \text{ and } \exists j : l_j(h') < l_j(h) \tag{4}$$

A Pareto Min-max solution means that a Pareto solution achieves the lowest loss on the maximal value in $\boldsymbol{l}(h)$. To obtain such a solution, there are research proposing to use the entropy function method to achieve min-max optimality and then continue to optimize all losses with projected gradient for Pareto optimality (Cui et al., 2021), which indicates that each objective cannot be further optimized without degrading others. Pareto min-max optimization maximizes the utility of the worst-performing objective.

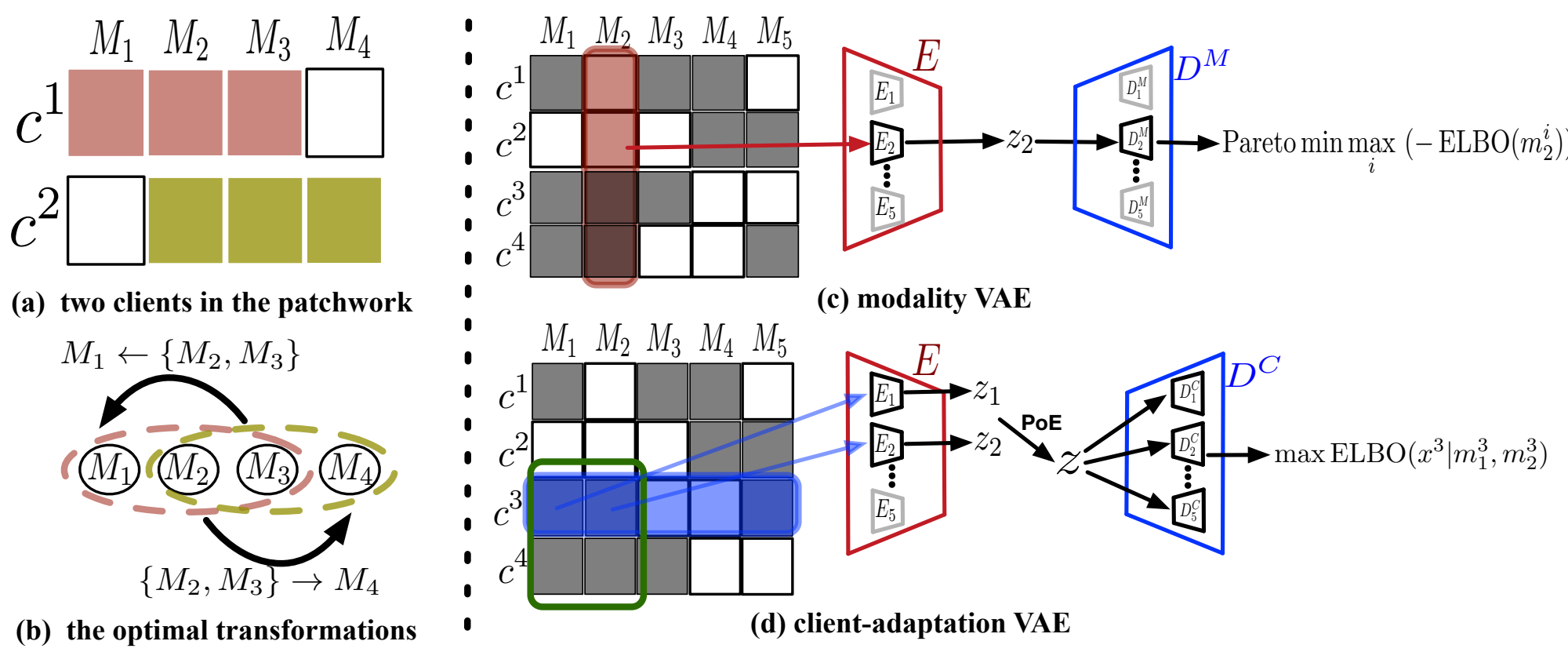

Figure 3: (a) and (b) illustrate the identification of the required dependencies for two clients. (a) the patchwork with two clients, where colored grids represent observed modalities and blank grids represent missing modalities; (b) the identified dependencies for the imputation. (c) and (d) illustrate our implementation framework `CLAP`. (c). modality VAE: each modality learns a VAE for obtaining the representation. The VAEs are trained by Pareto min-max optimization. (d). client-adaptation VAE (CA-VAE): CA-VAE shares the encoder ($E$) with modality VAE and has an extra decoder $D^C$.

### 4.2 An Overview of the Realization of `CLAP`

From Figure 3 (c) and (d), our framework consists of modality VAE and client-adaptation VAE, which synergistically address the three key challenges stated in Sec. 3.2.

**Modality VAE.** From Figure 3 (c), we learn a VAE for each modality $M_i$ to obtain the representation $z_i$, which has the encoder $E := \{E_1, ..., E_K\}$ and the decoder $D^M := \{D_1^M, ..., D_K^M\}$.

For the training of each modality VAE ($E_i$ and $D_i^M$), the data of the $i$-th modality $M_i$ come from multiple clients (e.g., $m_i^1, ..., m_i^N$) and are non-i.i.d. An averaged loss on all data may harm particular clients. The learned modality VAE could generate biased representation during imputation.

As shown in Figure 3 (c), to mitigate the statistical heterogeneity of the learned representation, we share the encoder for all clients. Meanwhile, we propose to balance the KL divergence of the learned representations by maximizing the client with the maximal KL divergence, so that each modality VAE encodes various data distributions of all clients to a common latent space, which we call ***vertical Pareto min-max optimization***.

**Client-Adaptation VAE (CA-VAE).** From Figure 3 (d), we learn a CA-VAE to model the dependencies among modalities, which shares the parameters of the encoder with modality VAE $E := \{E_1, ..., E_K\}$ and has an additional decoder $D^C := \{D_1^C, ..., D_K^C\}$. By using the two decoders ($D^M$ and $D^C$) separately, we relieve the potential conflicts when rebuilding the unimodal data in modality VAEs and multimodal data in CA-VAE.

[1]More details about the computation of $q(z|x^s)$ could be found in (Wu & Goodman, 2018)

To model the dependencies between modalities, we use the subset of the observed modalities $X'^i \subset X^i$ to infer other modalities and $X'^i$ itself. [2] Specifically, the subsets are identified as the intersection of the observed modalities between $X^i$ and other clients. For the example shown in Figure 3 (d), we use the modality combination $X^3 \cap X^4 = \{M_1, M_2\}$ to rebuild $X^3$. Then we use PoE to aggregate the representations $z_1 = h_{\theta_1}(m_1^3)$ and $z_2 = h_{\theta_2}(m_2^3)$, where $h_{\theta_i}$ denotes the encoder $E_i$ in Figure 3 (d). The aggregated $z$ is input to $D^C$ to rebuild all observed modalities $X^3$.

**The optimization of CA-VAE.** While a poor transformation learned from one client could worsen the imputation of multiple clients, to overcome the combination vulnerability, we propose ***horizontal Pareto min-max optimization*** to balance the dependency learning across the diverse combinations.

**The training of `CLAP`.** We train CLAP following the framework in federated learning (McMahan et al., 2017). Only the sum of the gradients is allowed to be transferred between the server and clients. We summarize the algorithm for the training of CLAP in Algorithm 1 in Appendix. The Pareto min-max optimization and other implementation details could also be found in Appendix because of the page limit.

### 4.3 Modality VAE and Vertical Pareto Min-Max Optimization

Multimodal data $\left\{\{m_j^i\}_{j=1}^K\right\}_{i=1}^N$ mostly have different dimensions when computing. To enhance the scalability of the imputation framework, we learn a modality VAE for each modality, which encodes $m_j^i$ to a latent representation $z = h_{\theta_j}(m_j^i)$ with fixed dimension, i.e.,

$$\min_{\theta_j,\phi_j} -(\mathbb{E}_{q_{\theta_j}(z|m_j^i)}\left[\log p_{\phi_j}(m_j^i|z)\right] - D_{KL}\left(q_{\theta_j}(z|m_j^i), p(z)\right)) \; where\; i = 1, 2, ..., N. \quad (5)$$

**Statistical heterogeneity and vertical Pareto min-max optimization.** From the equation (5), the KL divergence measures the difference between the distribution of the learned representations $q_{\theta_j}(z|m_j^i)$ and the spherical Gaussian $p(z)$. The presence of a diverse KL divergence among local clients suggests a significant statistical heterogeneity.

We propose vertical Pareto min-max optimization to address the statistical heterogeneity. Specifically, since there is a conflict when a global model minimizes all KL divergences, minimizing the maximal KL divergence on the worst client can increase the KL divergences of other clients, leading to more uniform KL divergences across all clients. Thus, the maximum KL divergence is an indicator of the uniformity of the KL divergences. We propose to enhance the worst client with minimal ELBO to minimize the maximal KL divergence.

$$\text{Pareto} \min_{\theta_j,\phi_j} \max_i -(\mathbb{E}_{q_{\theta_j}(z|m_j^i)}\left[\log p_{\phi_j}(m_j^i|z)\right] - D_{KL}(q_{\theta_j}(z|m_j^i), p(z))), m_j^i \; is\; observable. \quad (6)$$

### 4.4 Client-Apatation VAE and Horizontal Pareto Min-max Optimization

**Modality combination heterogeneity and CA-VAE.** Given a patchwork to be completed, the trainable transformations are defined. For a client $x^i$ with $k$ observed modalities, there are $2^k$ subsets and $2^k$ corresponding transformations[3]. While learning all transformations could have an exponential computing complexity, a natural problem is *how to identify the optimal transformations efficiently*.

**Identify the required dependencies on two clients.** To answer the above problem, we first consider there are two clients $c^1$ and $c^2$ with the modality combinations $X^1$ and $X^2$ shown in Figure 3 (a).

Both clients learn the dependencies on their observed modalities for the imputation of the other. From Figure 3 (a), to impute $m_1^2$ for $c^2$, $c^1$ could learn the transformation $M_2 \rightarrow M_1$ (or $M_3 \rightarrow M_1$) using its data $x^1$. However, the optimal dependencies for $c^2$ may be $\{M_2, M_3\} \rightarrow M_1$, which uses the intersection of $X^1$ and $X^2$ to infer other modalities. By using the intersection, the dependencies learned from $X^1$ maximize the utilization of the observed data in $X^2$ when imputing the missing modalities for $c^2$.

From the above analysis, the identified optimal dependencies learned by transformations shown in Figure 3 (b) are as follows.

$$\text{for } c^1\text{: } X^1 \cap X^2 \rightarrow X^1 \setminus X^2, \quad \text{for } c^2\text{: } X^2 \cap X^1 \rightarrow X^2 \setminus X^1 \quad (7)$$

[2]We also rebuild $X'^i$ for the rationality of the encoder.

[3]For each subset $X'^i \subset X^i$, there is a corresponding transformation $X'^i \rightarrow X^i \setminus X'^i$.

where $X^1 \setminus X^2$ denotes $X^1$ removes the elements in $X^2$. For two clients with different modality combinations, the identified dependencies mitigate the modality combination heterogeneity by using all common modalities, which is the intersection $X^1 \cap X^2$. This improves the transferability of the learned dependencies when applied to the target clients with different combinations.

**Identify the required dependencies on multiple clients.** For multiple clients, we identify the dependencies to be modeled according to Eq.(7) for each pair of clients. The required dependencies to be modeled by each client $c^i$ are formulated as follows.

$$for\ c^i:\ X^i \cap X^j \to X^i \setminus X^j,\ \forall j \in \{1, 2, ..., N\}\ ,\ j \neq i. \tag{8}$$

**Learning the identified dependencies by CA-VAE.** We consider the multimodal data $\left\{m_j^i\right\}_{j=1}^K$ in each client is generated by an unknown random process with a latent representation $z$. To learn the dependencies formulated in Eq.(8), we propose to approximate $X^i$ using $X^i \cap X^j$. We use $x^{i \cap j}$ to denote the subset of the variable $x^i$, which only contains the modality combination $X^i \cap X^j$. We optimize the KL-divergence between the learned and the true posterior distribution ($p\left(z|x^i\right)$) to model the dependencies in Eq.(8:

$$\min_\theta D_{KL}\left(q_\theta\left(z|x^{i\cap j}\right) \| p\left(z|x^i\right)\right), \tag{9}$$

Since $p\left(z|x^i\right)$ in Eq.(9) is intractable, we optimize Eq.(9) by giving the following proposition.

**Proposition 2.** *(proof in Appendix.) Eq.(10) formulates an appropriate ELBO, and minimizing Eq.(10) minimizes the KL-divergence formulated in Eq.(9), i.e.,*

$$\mathcal{L}_j^i\left(\theta, \phi; x^i\right) := -\left(\mathbb{E}_{q_\theta(z|x^{i\cap j})}\left[\log\left(p_\phi(x^i|z)\right] - D_{KL}\left(q_\theta\left(z|x^{i\cap j}\right) \| p(z)\right)\right), \tag{10}$$

where $\theta$ denotes the parameters of the encoder $E$ and $\phi$ denotes the parameters of the decoder $D^C$ shown in Figure 3 (d). $q_\theta\left(z|x^{i\cap j}\right)$ is calculated using PoE according to Eq.(3). $p(z)$ is the standard normal distribution.

For client $c^i$, we approximate the dependencies in Eq.(8) by optimizing the sum of the KL-divergence between $q(z|x^{i\cap j})$ and $p(z|x^i)$, where $j$ is the selected other clients with different modality conbinations ($X^i \neq X^j$).

$$\min_{\theta,\phi} \mathcal{L}^i\left(\theta, \phi; x^i\right) := \sum_j \mathcal{L}_j^i\left(\theta, \phi; x^i\right). \tag{11}$$

**Modality combination vulnerability and horizontal Pareto min-max optimization.** While a poor transformation learned from one client could worsen the imputation of multiple clients. We propose to mitigate the worst-case effect by Pareto min-max optimization, i.e.,

$$\text{Pareto}\ \min_{\theta,\phi} \max_i\ \mathcal{L}^i\left(\theta, \phi; x^i\right). \tag{12}$$

## 5 EXPERIMENTS

**Baselines.** As we tackle a new problem and there is no existing method that focuses on it, we modify the training scheme of existing methods on the modality imputation using an existing distributed learning scheme (e.g., FedAvg (McMahan et al., 2017)) as ours. In particular, in each communication round, each client trains its local model using the observed modalities, then the server averages the model weights to update the global model. The baselines include MVAE (Wu & Goodman, 2018), MMVAE (Shi et al., 2019) and MoPoE-VAE (Sutter et al., 2020b).



MVAE MMVAE MoPoE **CLAP**

Figure 4: Imputation results of all methods on PolyMNIST. Each column represents a sample. Red boxes indicate the missing modalities.

**Datasets.** Following the work (Sutter et al., 2020b), we evaluate our method with baselines on benchmark datasets with various modalities, including PolyMNIST, MNIST-SVHN-TEXT, CelebA

and CUB. More importantly, the practicability of our method is validated on a real-world clinical distributed dataset eICU (Pollard et al., 2018).

**Evaluation metrics.** We evaluate the performance of all methods from two aspects. **1. generation quality;** the models take the observed modalities as inputs to generate missing modalities. We assess the coherence of generated modalities with pre-trained classifiers to evaluate whether the imputed samples could be classified correctly. Meanwhile, we also approximate the log-likelihoods of the output data to reflect the generation quality. For the eICU dataset, which is tabular data and has many modalities, we report the mean square error (MSE) between the generated results and the ground truth. **2. representation quality;** the latent space of the data is supposed to be informative. We build the downstream tasks by using the learned classifiers to measure the quality of learned representations. All experiments are run 5 times to calculate the average results with stds. More experimental results, more ablation studies and discussions could be found in Appendix.

### 5.1 Experiments on PolyMNIST

The PolyMNIST dataset (Sutter et al., 2020b) is synthesized with the MNIST dataset. Each sample contains 5 image modalities with different backgrounds and styles with the same digit label. There are 60,000 samples in the training set and 10,000 samples in the test set. We create the distributed environment by partitioning the dataset into 5 clients. Each client contains 6 classes of samples. Different clients involve different classes of data. In each client, we mask less than 2 modalities at random.

Table 1: Imputation coherence accuracy (%) of 5 modalities on the PolyMNIST dataset.

| Model | $M_1$ | $M_2$ | $M_3$ | $M_4$ | $M_5$ |
|---|---|---|---|---|---|
| MVAE | $22.1_{\pm 1.1}$ | $19.5_{\pm 0.5}$ | $15.2_{\pm 1.4}$ | $68.7_{\pm 1.2}$ | $19.6_{\pm 1.0}$ |
| MMVAE | $36.3_{\pm 0.1}$ | $60.5_{\pm 0.4}$ | $61.3_{\pm 1.0}$ | $64.4_{\pm 2.0}$ | $34.4_{\pm 0.8}$ |
| MoPoE | $38.5_{\pm 0.6}$ | $69.4_{\pm 1.4}$ | $72.4_{\pm 1.4}$ | $75.7_{\pm 0.5}$ | $45.6_{\pm 1.0}$ |
| CLAP | $\mathbf{45.2}_{\pm 0.2}$ | $\mathbf{69.4}_{\pm 1.1}$ | $\mathbf{74.5}_{\pm 0.1}$ | $\mathbf{79.3}_{\pm 0.8}$ | $\mathbf{50.5}_{\pm 0.3}$ |

Table 2: Linear classification accuracy (%) on the representation of the modality subsets on the PolyMNIST dataset.

| Model | $\{M_4\}$ | $\{M_4, M_5\}$ | $\{M_2, M_4, M_5\}$ | Average |
|---|---|---|---|---|
| MVAE | $81.4_{\pm 0.4}$ | $75.7_{\pm 0.9}$ | $76.7_{\pm 0.2}$ | $77.9_{\pm 0.9}$ |
| MMVAE | $86.9_{\pm 1.1}$ | $78.9_{\pm 0.5}$ | $83.2_{\pm 1.0}$ | $83.0_{\pm 1.1}$ |
| MoPoE | $79.2_{\pm 0.4}$ | $80.1_{\pm 1.5}$ | $\mathbf{97.9}_{\pm 0.9}$ | $85.7_{\pm 1.5}$ |
| CLAP | $\mathbf{87.6}_{\pm 0.2}$ | $\mathbf{91.8}_{\pm 0.4}$ | $95.9_{\pm 0.2}$ | $\mathbf{91.9}_{\pm 0.4}$ |

The experimental results on PolyMNIST dataset are shown in Table 1, and 2. From Table 1, MVAE has a worse imputation coherence accuracy because it fails to learn the dependencies among modalities. MoPoE achieves better imputation coherence than other baselines as it learns all transformations among modalities. CLAP achieves the best imputation coherence as it models the required dependencies for imputation, which mitigates the potential modality combination vulnerability. For the quality of the learned representation shown in Table 2, CLAP still maintains the best accuracy as our method addresses the heterogeneity by learning a common latent representation for all clients. We visualize the generated result in Figure 4. From Figure 4, the baseline methods are easily confused by the images '7' and '9', and CLAP correctly impute the missing modalities.

### 5.2 Experiments on More Complicated Dataset Bimodal CelebA

**Bimodal CelebA** The Bimodal CelebA dataset (Sutter et al., 2020a) is extended from CelebA dataset (Liu et al., 2015). Each face image is attached with a textual modality. It is generated with the 40 attributes describing the face. Compared with the MNIST-SVHN-TEXT dataset, the face images in the Bimodal CelebA dataset are more complex. Also, the text strings are longer and more complicated. We split the dataset into 5 clients. For the setting of statistical heterogeneity, we set different portions of *Male* samples in local clients.

The results of imputation coherence and latent space classification are shown in Table 3. CLAP surpasses all baselines and achieves the highest imputation coherence and latent space classification accuracy.

Table 3: Imputation coherence and latent space classification results of 2 modalities $I$ (Image) and $T$ (Text) on the Bimodal CelebA dataset. The mean average precision (mAP) (%) is averaged over all 40 attributes.

| Model | Coherence | | Latent Classification | |
|---|---|---|---|---|
| | I | T | I | T |
| MVAE | $35.5_{\pm 0.1}$ | $34.7_{\pm 0.4}$ | $16.0_{\pm 0.3}$ | $7.1_{\pm 0.2}$ |
| MMVAE | $34.4_{\pm 0.2}$ | $32.7_{\pm 0.3}$ | $43.7_{\pm 0.2}$ | $41.5_{\pm 0.1}$ |
| MoPoE | $34.9_{\pm 0.3}$ | $33.7_{\pm 0.5}$ | $43.8_{\pm 0.1}$ | $41.2_{\pm 0.3}$ |
| CLAP | **37.4**$_{\pm 0.3}$ | **35.2**$_{\pm 0.1}$ | **44.2**$_{\pm 0.1}$ | **45.4**$_{\pm 0.3}$ |

Table 4: Results on eICU dataset. MSE and latent classification AUC (%) are averaged over 13 modalities and certain subsets. The log-likelihoods of joint distribution $X$ are reported.

| Model | MSE | Latent Classification | Likelihood |
|---|---|---|---|
| MVAE | $0.0661_{\pm 0.0002}$ | $56.3_{\pm 0.3}$ | $-341.1_{\pm 0.5}$ |
| MMVAE | $0.0681_{\pm 0.0004}$ | $53.0_{\pm 0.2}$ | $-340.4_{\pm 0.6}$ |
| MoPoE | - | - | - |
| CLAP | **0.0653**$_{\pm 0.0002}$ | **62.3**$_{\pm 0.2}$ | **-340.2**$_{\pm 0.7}$ |

### 5.3 Experiments on a Real-world Clinical Dataset eICU

To further confirm the practicability of CLAP, we use a real-world clinical dataset eICU (Pollard et al., 2018). eICU contains the records of the patients to ICUs with hospital information, and each hospital is a client (Cui et al., 2022). The data is preprocessed as in (Sheikhalishahi et al., 2019). We select 13 features (e.g., FiO2, O2 Saturation, etc.) as 13 modalities. Each instance has the 48-hour monitoring data of the 13 features. The label is binary indicating whether the patient is alive. The experimental results are shown in Table 4, in which MoPoE can not work on the eICU dataset due to the vast number ($2^{13}$) of modality subsets. We use AUC to evaluate the classification results due to the severe label imbalance (more than $90\%$ samples have negative labels). From Table 4, CLAP outperforms baselines in all three metrics.

### 5.4 Ablation Studies

**Vertical and horizontal Pareto min-max optimization.** We conduct ablation studies on PolyMNIST to validate the effect of Pareto min-max optimization. In particular, we use the averaged loss to replace Pareto min-max optimization in Eq.(6) and Eq.(12). The averaged results over all 5 modalities are shown in Table 5, where w/o both denotes that both losses are replaced with the averaged losses. w/o modality means that we only replace vertical Pareto min-max optimization in Eq.(6) with the averaged loss, and w/o client means that we only replace horizontal Pareto min-max optimization in Eq.(6) with the averaged loss. The results are shown in Table 5. The ablation studies on more clients could be found in Appendix.

Table 5: Ablation study on the PolyMNIST dataset. Coherence accuracy, log-likelihood and latent classification accuracy (%).

| Method | Coherence | Likelihood | Latent |
|---|---|---|---|
| w/o both | $56.4_{\pm 0.2}$ | $-6672_{\pm 1}$ | $53.3_{\pm 0.4}$ |
| w/o modality | $57.3_{\pm 0.2}$ | $-6642_{\pm 3}$ | $54.4_{\pm 0.1}$ |
| w/o client | $56.9_{\pm 0.1}$ | $-6633_{\pm 4}$ | $55.1_{\pm 0.2}$ |
| CLAP | **58.7**$_{\pm 0.2}$ | **-6603**$_{\pm 4}$ | **56.6**$_{\pm 0.1}$ |

## 6 Conclusion

In this paper, we propose a practical multimodal learning problem. Given an incomplete patchwork, we propose a scalable framework, CLAP, to achieve an efficient and valid imputation by jointly learning from the patchwork vertically and horizontally without direct access to the raw data. Empirical results on both benchmark and real-world medical data sets demonstrated the effectiveness, superiority, and practicability of our proposed method. We hope this work opens venues for future research, for example, large models in multimodal learning, unsupervised multimodal learning, etc.

## 7 Acknowledgments*

This work is funded by the Natural Science Fundation of China(NSFC. No. 62176132) and the Guoqiang Institute of Tsinghua University, with Grant No. 2020GQG0005.

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

## CONTENTS

## A THEORETICAL PROOFS

### A.1 PROOF OF PROPOSITION 1

**Assumption 3.** *All modalities form a connected graph in the patchwork.*

Firstly, we prove that if a patchwork satisfies Assumption 1 and Assumption 3, then the patchwork could be completed.

Suppose we use $f(M_i, M_j)$ to denote a transformation from $M_i$ to $M_j$. If there are a client $c^k$ owning the two modalities $\{M_i, M_j\} \subset X^k$, the transformation $f(M_i, M_j)$ could be optimized by learning from the data.

From Assumption 3, if the patchwork forms a connected graph, each pair of modalities has an undirected path from one modality to the other, i.e.,

$$\forall i, j \in \{1, 2, ..., K\}, \exists p,\ s.t.,\ M_i \rightarrow M_j, \tag{13}$$

where $p$ denotes a path on the graph.

As each edge denotes an optimized transformation, the path between $M_i$ and $M_j$ corresponds to a composed transformation $f(M_i, M_{k^1}) \circ f(M_i, M_{k^2}) ... \circ f(M_{k^m}, M_j)$. Therefore, each missing modality could be imputed by the optimized transformations and the patchwork could be completed.

Then we prove that if a patchwork could be completed by the optimized transformations, the patchwork satisfies Assumption 1 and Assumption 3.

If the patchwork violates Assumption 1 and there exists a client has no data, i.e.,

$$\exists i \in \{1, 2, ..., N\}, X^i = \emptyset, \tag{14}$$

then $c^i$ cannot impute its missing modalities and the patchwork cannot be completed. This violates the premise that the patchwork could be completed. Therefore, the patchwork satisfies Assumption 1.

If the patchwork violates Assumption 3, there exists two modalities $\left\{M^i, M^j\right\}$ and no path between them, i.e.,

$$\exists i, j \in \{1, 2, ..., K\}, \nexists p,\ s.t.,\ M_i \rightarrow M_j. \tag{15}$$

Since there is no edge between $M_i$ and $M_j$, there is no client that owns the two modalities $\{M_i, M_j\}$. If a client $c^l$ has the modality $M_i$ and it can impute $M_j$ from the learned transformations, this means that there exists a path between another observed modality $M_k$ and $M_j$, i.e.,

$$\exists M_k \in X^l, \exists p,\ s.t.,\ M_k \rightarrow M_j. \tag{16}$$

Meanwhile, $c^l$ has both modality $M_i$ and $M_k$, so there exists a path between $M_k$ and $M_i$.

$$\exists p,\ s.t.,\ M_k \rightarrow M_i. \tag{17}$$

Combining the above two Equations, there exists $p$ from $M_i$ to $M_j$ and this violates Eq.(15). Therefore, Assumption 3 holds.

If a client $c^l$ has the modality $M_j$ and it can impute $M_i$, we can prove Assumption 3 holds similarly.

If a client $c^l$ does not have $M_i$ and does not have $M_j$, $c^l$ can impute both modalities using the observed modalities. Then there exists a modality $M_k \in X^l$ and there exists a path between $M_k$ and $M_i$ and a path between $M_k$ and $M_j$. Therefore, there exists $p$ from $M_i$ to $M_j$. This violates Eq.(15) and Assumption 3 holds.

### A.2 Proof of Proposition 2

Firstly, we prove that Eq.(10) formulates an appropriate ELBO, i.e.,

$$\log p(x^i) \geq -\mathcal{L}^{i,j}(\theta, \phi, x^i). \tag{18}$$

We prove it as follows.

$$\log p(x^i) = D_{KL}\left(q_\theta(z|x^{i\cap j})||p(z|x^i)\right) + E_{q_\theta(z|x^{i\cap j})}\left[\log \frac{p(z, x^i)}{q_\theta(z|x^{i\cap j})}\right] \tag{19a}$$

$$\geq E_{q_\theta(z|x^{i\cap j})}\left[\log \frac{p(z, x^i)}{q_\theta(z|x^{i\cap j})}\right] \tag{19b}$$

$$= E_{q_\theta(z|x^{i\cap j})} \log p_\phi(x^i|z) - D_{KL}\left(q_\theta\left(z|x^{i\cap j}\right)\|p(z)\right) \tag{19c}$$

$$= -\mathcal{L}^{i,j}(\theta, \phi, x^i). \tag{19d}$$

Since $D_{KL}\left(q_\theta(z|x^{i\cap j})||p(z|x^i)\right) \geq 0$, the inequality Eq.(19b) holds.

Then we demonstrate that minimizing $\mathcal{L}^{i,j}(\theta, \phi, x^i, x^j)$ minimizes $D_{KL}\left(q_\theta\left(z|X^{i\cap j}\right)\|p\left(z|X^i\right)\right)$ in Eq.(9).

From Eq.(19a), $\log p(x^i)$ is pre-defined. Maximizing $E_{q_\theta(z|x^{i\cap j})}\left[\log \frac{p(z,x^i)}{q_\theta(z|x^{i\cap j})}\right]$ minimizes $D_{KL}\left(q_\theta(z|x^{i\cap j})||p(z|x^i)\right)$. In Eq.(19a), $p(z, x^i)$ is agnostic, so we use $p_\phi(x^i|z)p(z)$ to approximate $p(z, x^i)$. Therefore, $-\mathcal{L}^{i,j}(\theta, \phi, x^i)$ is an appropriate approximation of $E_{q_\theta(z|x^{i\cap j})}\left[\log \frac{p(z,x^i)}{q_\theta(z|x^{i\cap j})}\right]$, and maximizing $-\mathcal{L}^{i,j}(\theta, \phi, x^i)$ minimizes $D_{KL}\left(q_\theta\left(z|x^{i\cap j}\right)\|p\left(z|x^i\right)\right)$.

## B Convergence Analysis

The work (Cui et al., 2021) provides the convergence analysis of Pareto min-max optimization in section C in Appendix. We present the analysis in the following.

The convergence of Pareto min-max optimization. (More proof details could be found in (Cui et al., 2021).) Recall that Pareto min-max optimization aims to achieve a Pareto solution $h^*$ that has the lowest loss on the maximal value in $\boldsymbol{l}(h) = [l_1(h), l_2(h), ..., l_N(h)]$. Pareto min-max optimization obtains such a $h^*$ by a two-staged constrained multi-objective optimization.

**The convergence of the first stage**: Suppose we use $\theta^t$ and $\delta^t$ to denote the parameters of $h$ and the parameter $\delta$ in the $t$-th iteration, (Cui et al., 2021) proves that in each iteration, $\hat{l}_{max}(h_{\theta^t}, \delta^t)$ decreases in Lemma 1 in the Appendix.

$$\begin{gathered}\hat{l}_{\max}\left(h_{\theta^{t+1}}, \delta_l^t\right) \leq \hat{l}_{\max}\left(h_{\theta^t}, \delta_l^t\right) \\ \hat{l}_{\max}\left(h_{\theta^t}, \delta_l^{t+1}\right) \leq \hat{l}_{\max}\left(h_{\theta^t}, \delta_l^t\right) \\ \hat{l}_{\max}\left(h_{\theta^{t+1}}, \delta_l^{t+1}\right) \leq \hat{l}_{\max}\left(h_{\theta^t}, \delta_l^t\right)\end{gathered}$$

As $\hat{l}_{max}(h_{\theta^t}, \delta^t) > 0$ and decreases in each iteration, Pareto min-max optimization achieves min-max optimality and the algorithm converges.

**The convergence of the second stage**: (Cui et al., 2021) proves that in each iteration, the objective $l_i(h_\theta^t)$ $(i = 1, 2, ..., N)$ decreases or remains unchanged.

$$l_i\left(h_{\theta^{t+1}}\right) \leq l_i\left(h_{\theta^t}\right), \forall i \in \{1, ., N\}$$

This means that the objective in the second stage ($\frac{1}{N}\sum_{i=1}^{N} l_i(h)$) decreases or remains unchanged without violating the constraints. Finally, Pareto min-max optimization achieves Pareto optimality and the algorithm converges.

**The convergence is not affected by the statistical and modality heterogeneity**. From the above analysis, the convergence of Pareto min-max optimization is guaranteed from the perspective of constrained multi-objective optimization.

**Algorithm 1** The training of CLAP

**Input:** the number of client $N$, the number of modalities $K$, epoch $T_m$, batch size $B_m$, initialization of the encoder $E$ ($\theta^k$), the decoder $D^M$ ($\phi_k^M$) the decoder $D^C$ ($\phi_k^C$), where $k = 1, 2, ..., K$.

**for** $t = 0, ..., T_m - 1$ **do**
  randomly selects a subset of clients $S_t$
  *the training of modality VAEs*
  **for** modality $M_k, k = 1, 2, ..., K$ **do**
    **for** client $c^i \in S_t$ in parallel **do**
      (if $M_k$ is observed) draw mini-batch $m_k^i \sim X^i$
      calculate the ELBO: $\text{ELBO}(m_k^i)$
      then calculate the gradients with respect to parameters $\theta^k$ $\phi_k^M$;
    **end for**
    **Server** aggregates the gradients of selected clients and update the parameters $\theta^k$ $\phi_k^M$ by Pareto min-max optimization;
  **end for**
  *the training of CA-VAEs*
  **for** client $c^i \in S_t$ in parallel **do**
    draw mini-batch $x^i \sim X^i$
    calculate the losses $L_j^i(\theta, \phi, x^i)$ ($c^j \in S_t$) according to Eq.(10)
    then calculate the gradients of losses for parameters $\theta^k$ $\phi_k^C$ ($k = 1, 2, ..., K$);
  **end for**
  **Server** aggregates the gradients of selected clients and update the parameters $\theta^k$ and $\phi_k^C$ ($k = 1, 2, ..., K$) by Pareto min-max optimization.
**end for**
**Output:** the learned model $\theta^k$, $\phi_k^M$, $\phi_k^C$ ($k = 1, 2, ...., K$).

**statistical heterogeneity**: Pareto min-max optimization is applied to mitigate the statistical heterogeneity. Even if there is severe statistical heterogeneity in local clients, Pareto min-max optimization still converges to a Pareto stationary solution, whose gradient is close to zero.

**modality heterogeneity**: Different modality combinations correspond to different objectives. Modality heterogeneity may induce more objectives to be optimized. During the Pareto min-max optimization, the objectives decrease in each iteration and the algorithm still converges after $T$ iterations.

# C Algorithm and Optimization

## C.1 Algorithm

We summarize the algorithm for the training of CLAP in Algorithm 1. For the inference of CLAP, we present the process of imputation in Algorighm 2. During training, we use the intersection of observed modalities ($X^i \cap X^j$) between modalities to rebuild all observed modalities. Note that each client has $N - 1$ modality intersection with other clients. When aggregating the gradients, in each client, only the parameter gradients of encoders and decoders corresponding to visible modalities are transmitted. In the inference, we use the intersection which has the most modalities to infer the missing modalities. This corresponds to Line 5 in Algorithm 2.

## C.2 Pareto Min-Max Optimization

Pareto min-max optimization (Cui et al., 2021) aims to achieve a Pareto solution $h^*$ that has the lowest loss on the maximal value in $\boldsymbol{l}(h) = [l_1(h), l_2(h), ..., l_N(h)]$. Pareto min-max optimization obtains such a $h^*$ by a two-staged constrained multi-objective optimization.

**Algorithm 2** The inference of `CLAP`

**Input:** the patchwork to be completed, the number of clients $N$, the number of modalities $K$, the learned model $\theta^k$, $\phi_k^M$, $\phi_k^C$ $(k = 1, 2, ...., K)$.

1: **for** $i = 1, 2, ..., N$ **do**
2:     **for** $k = 1, 2, ..., K$ **do**
3:         (if $m_k^i$ is observed) calculate the latent representation $z$ using the encoder $\theta^k$;
4:     **end for**
5:     calculates the representation $z$ using the maximal intersection modality combination by PoE;
6:     rebuild all missing modalities using the decoder $D^C$ $(\phi_k^C,\ k = 1, 2, ..., K)$;
7: **end for**
8: **Output:** the completed patchwork.

**The first stage**: it achieves a min-max solution $h^1$ by entropy function method, which optimizes a surrogate function defined as follows

$$\hat{l}_{max}\left(h, \delta\right) = \delta \ln \sum_{i=1}^{N} \exp\left(\frac{l_i(h)}{\delta}\right).$$

$\hat{l}_{max}\left(h, \delta\right)$ satisfies that

$$l_{max}(h) \leq \hat{l}_{max}\left(h, \delta_l\right) \leq l_{max}(h) + \delta \ln(N),$$

where $\delta > 0$ is a hyper-parameter. It will decrease in each iteration.

**The second stage**: it continues to optimize the obtained solution $h^1$ to achieve Pareto optimality with constrained Pareto optimization. The objective to be optimized is as follows:

$$\min_h \frac{1}{N} \sum_{i=1}^{N} l_i(h), s.t., l_i(h) \leq l_i(h^1), \forall i \in \{1, 2, ..., N\}.$$

In this way, Pareto min-max optimization achieves a Pareto solution, in which each objective cannot be further optimized without degrading others. Meanwhile, it maintains its min-max optimality as it achieves the minimal maximum loss value.

### C.3 Discussions on Limitations

This paper focuses on generating missing modalities for local clients. However, privacy and communication concerns arise when learning generative models collaboratively, and `CLAP` is no exception.

**Privacy.** The implementation of `CLAP` involves collaboratively learning generative VAEs for imputing missing modalities. The transmission of gradient information between the server and clients may result in information leakage. To address this privacy concern, differential privacy may be useful to protect privacy from leakage during the training of the generative models.

**Communication.** Generative models typically have more parameters compared to classification models, which may result in more communication and computation overhead for local clients. In this regard, transferring the most valuable gradient information for model training is a potential research direction to reduce communication overhead.

## D Implementation

### D.1 Evaluation

In patchwork learning, models learn to project the samples with missing modalities into a joint latent representation and reconstruct the complete modalities. We evaluate the performance of all methods from two aspects: generation quality and representation quality. Generation quality can be assessed by generation coherence and Test Set Log-likelihood. For the eICU dataset, we also report the mean square error (MSE) between generated and input samples. Representation quality can be assessed by

linear classifier accuracy in latent space. The sample labels refer to digit numbers in the PolyMNIST dataset and the MNIST-SVHN-TEXT dataset, 40 binary attributes in the Bimodal CelebA dataset, and mortality in the eICU dataset. The results are averaged over all clients.

**Generation Coherence**. A classifier is trained for each modality to predict the label of samples. During the evaluation of multimodal methods, all the modalities are reconstructed from incomplete modalities. Then we predict labels from the generated modalities and calculate accuracy (for the PolyMNIST dataset and the MNIST-SVHN-TEXT dataset), mean average precision (for the Bimodal CelebA dataset), or AUC (for the eICU dataset) using the true labels.

**Test Set Log-likelihood**. We generate full modalities for each sample and estimate the test set log-likelihood with 15 importance samples as in (Sutter et al., 2020b).

**Linear Classifier Accuracy in Latent Space**. Patchwork learning methods project samples into a joint latent representation and total modalities can be reconstructed from the latent representation. The learned latent space of quality should be constructed and contain intact information of all the modalities even if the input modalities are incomplete. Therefore, we perform linear classification in the latent space to predict the sample label. A logistic regression classifier is trained with 500 samples in the training set for each modality subset in each client. And it is tested in the latent space of test sets.

### D.2 More Experiments

**MNIST-SVHN-TEXT** dataset (Sutter et al., 2020a) is created based on MNIST-SVHN dataset (Shi et al., 2019), which is composed of image modalities MNIST and SVHN, as well as a text modality. In particular, MNIST (Deng, 2012) modality is a handwritten digit image. SVHN (Netzer et al., 2011) modality is an image of a digit cropped from street view pictures. Each sample has the same label for all images. The text modality is an English string indicating the digit label. Each MNIST image pairs with 20 SVHN images. The dataset has 20×60,000 training samples and 20×10,000 test samples. For the setting of data heterogeneity, we partition the dataset into 10 clients, and each client contains 6 classes of samples. In each client, there is less than one modality to be masked.

The performance of all methods on this dataset is shown in Table 6. From Table 6, our method outperforms all baselines and achieves the best coherence. The modeling of the SVHN modality is mediocre as this modality is complex and noisy. `CLAP` still maintains the best coherence (58.9). As there may exist a trade-off between coherence and log-likelihood, our method achieves a comparable log-likelihood with baselines. More results are shown in Appendix.

Table 6: Imputation coherence accuracy (%) of 3 modalities $M$ (MNIST), $S$ (SVHN) and $T$ (Text) on the MNIST-SVHN-TEXT dataset. The log-likelihoods are tested on $X = \{M, S, T\}$

| Model | Coherence | | | Likelihood |
|---|---|---|---|---|
| | $M$ | $S$ | $T$ | $X$ |
| MVAE | $75.5_{\pm 0.3}$ | $47.7_{\pm 1.4}$ | $80.4_{\pm 0.7}$ | $-1971_{\pm 3}$ |
| MMVAE | $88.6_{\pm 0.6}$ | $21.5_{\pm 2.1}$ | $76.8_{\pm 1.4}$ | $-1838_{\pm 2}$ |
| MoPoE | $89.4_{\pm 0.7}$ | $54.4_{\pm 0.2}$ | $92.1_{\pm 0.3}$ | $-1840_{\pm 1}$ |
| `CLAP` | $\mathbf{95.3}_{\pm 0.8}$ | $\mathbf{58.9}_{\pm 0.3}$ | $\mathbf{92.3}_{\pm 0.3}$ | $\mathbf{-1838}_{\pm 1}$ |

**CUB.** We also conduct our experiments on CUB datasets. CUB Hwang et al. (2021) contains 11,788 photos of 200 kinds of birds in natural scenes, each annotated with 10 fine-grained captions describing the bird's appearance characteristics collected through Amazon Mechanical Turk (AMT). We partition the data into 10 clients. Each client contains 90 different kinds of birds. For the two modalities (image and text), we randomly mask one modality for the randomly selected clients. The results of MSE, linear classification, and log-likelihood are shown in Table 7, where I and T stand for image and text modalities respectively. As shown in the table, CLAP still outperforms all baselines on this complex dataset.

**PolyMNIST.** The log-likelihood results are reported in Table 8, which evaluates the generation quality. MVAE has a worse imputation coherence accuracy because it fails to learn the dependencies among modalities. MoPoE achieves better imputation coherence than other baselines as it learns all transformations among modalities. `CLAP` achieves the best log-likelihoods as it models the required dependencies for imputation compared with baselines.

Table 7: Experimental results on CUB

| method | MSE | Latent Classification | | | Likelihood | | |
|---|---|---|---|---|---|---|---|
| | I | {I} | {T} | {I, T} | I | T | X |
| MVAE | 0.0298 | 13.4 | 1.7 | 5.3 | -6437 | -156 | -6549 |
| MMVAE | 0.0425 | 13.8 | 2.9 | 3.5 | -6650 | **-117** | -6745 |
| MoPoE | 0.0241 | 13.0 | 3.5 | 4.9 | -6283 | -148 | -6374 |
| CLAP | **0.0233** | **15.3** | **3.5** | **5.7** | **-6251** | -142 | **-6342** |

Table 8: The log-likelihoods of 5 modalities and the joint log-likelihoods of $X = \{M_i, i = 1, 2, ...5\}$ on PolyMNIST dataset.

| Model | $M_1$ | $M_2$ | $M_3$ | $M_4$ | $M_5$ | $X$ |
|---|---|---|---|---|---|---|
| MVAE | -1495$_{\pm 2}$ | -1292$_{\pm 1}$ | -1252$_{\pm 4}$ | -1373$_{\pm 2}$ | -1431$_{\pm 1}$ | -6849$_{\pm 1}$ |
| MMVAE | -1347$_{\pm 1}$ | **-1199**$_{\pm 5}$ | -1274$_{\pm 4}$ | -1460$_{\pm 1}$ | -1346$_{\pm 2}$ | -6964$_{\pm 4}$ |
| MoPoE | -1412$_{\pm 3}$ | -1237$_{\pm 2}$ | -1201$_{\pm 2}$ | -1494$_{\pm 4}$ | **-1312**$_{\pm 6}$ | -6715$_{\pm 5}$ |
| CLAP | **-1319**$_{\pm 2}$ | -1207$_{\pm 1}$ | **-1177**$_{\pm 3}$ | **-1353**$_{\pm 2}$ | -1349$_{\pm 3}$ | **-6603**$_{\pm 4}$ |

**MNIST-SVHN-TEXT.** The results of linear classification accuracy on MNIST-SVHN-TEXT are in Table 9.

Table 9: Linear classification accuracy (%) in latent space of some modality subsets on the MNIST-SVHN-TEXT dataset. The average accuracy of all possible subsets is also reported.

| Model | $\{S\}$ | $\{M, S\}$ | $\{M, S, T\}$ | Average |
|---|---|---|---|---|
| MVAE | 32.7$_{\pm 0.2}$ | 66.5$_{\pm 1.3}$ | 48.8$_{\pm 0.1}$ | 71.1$_{\pm 0.5}$ |
| MMVAE | 79.7$_{\pm 0.5}$ | 65.0$_{\pm 0.8}$ | 47.2$_{\pm 0.7}$ | 79.1$_{\pm 0.2}$ |
| MoPoE | **79.9**$_{\pm 0.3}$ | 68.5$_{\pm 0.3}$ | 49.0$_{\pm 0.2}$ | 80.5$_{\pm 0.2}$ |
| CLAP | 79.4$_{\pm 0.1}$ | **68.7**$_{\pm 0.6}$ | **49.2**$_{\pm 0.3}$ | **80.6**$_{\pm 0.1}$ |

**eICU.** In the original setting, there are 13 modalities in the eICU dataset, which means the MoPoE can not work due to the vast number ($2^{13}$) of modality subsets. To make the comparison feasible, we discard the last modality and divide the rest 12 modalities into 4 modality groups. The results are shown in the Table 10. In this setting, our method also outperforms all the baselines on all metrics.

### D.3 Training Details

We use Adam optimizer for training and the training batch size of all experiments is set as 256. The architecture of classifiers for the coherence test is the same as the unimodal encoder architecture except for the last layer. The model architecture and training hyperparameters are the same for different methods. When calculating the training loss, the likelihood of reconstructed modalities is weighted according to their data dimensions.

**Bimodal CelebA.** The learning rate is set as 0.0005, $\beta$ is set as 1.0, dimension of latent vectors is set as 64. The models are trained for 200 epochs. The encoders and decoders for both the Image and Text modalities are based on residual convolutional blocks. We use modality-specific latent space as in (Daunhawer et al., 2020) to improve the generation quality. The modality-specific latent vector occupies half of the latent vectors.

**PolyMNIST**. The learning rate is set as 0.001, $\beta$ is set as 0.4, dimension of latent vectors is set as 512. The models are trained for 300 epochs. The unimodal encoders are convolutional neural networks and the decoders are based on transposed convolutional layers.

**CUB**. The learning rate is set as 0.0005, $\beta$ is set as 0.4, dimension of latent vectors is set as 64. The models are trained for 75 epochs. The encoders for both the Image and Text modality are based on convolutional neural networks and the decoders are based on transposed convolutional layers.

**MNIST-SVHN-TEXT**. The learning rate is set as 0.001, $\beta$ is set as 0.4, dimension of latent vectors is set as 20. The models are trained for 150 epochs. The encoder and decoder of the MNIST modality are fully-connected neural networks. For the SVHN and TEXT modality, they are convolutional and transposed convolutional neural networks.

Table 10: Results on eICU dataset with 4 modalities.

| Method | MSE | | | | Latent Classification | Likelihood |
|---|---|---|---|---|---|---|
| | $M_1$ | $M_2$ | $M_3$ | $M_4$ | Average | $X$ |
| MVAE | 0.0366 | 0.0426 | 0.1006 | 0.0876 | 40.2 | -319 |
| MMVAE | 0.0380 | 0.0440 | 0.1029 | 0.0928 | 38.3 | -320 |
| MoPoE | 0.0360 | 0.0426 | 0.0997 | 0.0848 | 41.5 | -319 |
| CLAP | **0.0360** | **0.0422** | **0.0997** | **0.0847** | **43.3** | **-316** |

**eICU**. The learning rate is set as 0.0005, $\beta$ is set as 2.0, dimension of latent vectors is set as 32. The models are trained for 150 epochs. The encoders are composed of LSTM and fully connected layers. And the decoders are fully-connected neural networks.

# E More Ablation Studies

## E.1 More Clients

We conduct experiments on the PolyMNIST dataset with more clients (50 and 100). The results are shown in Table 15. When we split the dataset into more clients, the statistical heterogeneity gets more complex. Thus the latent classification results and log-likelihood decline for all the models and our method still outperforms the baselines.

Table 11: Latent space classification and log-likelihood results of some modality subsets on the PolyMNIST dataset. The average accuracy of all possible subsets.

| Client Number | Model | Latent Classification | | | | Likelihood | | | |
|---|---|---|---|---|---|---|---|---|---|
| | | $\{M_4\}$ | $\{M_4, M_5\}$ | $\{M_2, M_4, M_5\}$ | Average | $M_2$ | $M_4$ | $M_5$ | $X$ |
| 50 | MVAE | 34.5 | 31.5 | 30.7 | 21 | -1268 | -1518 | -1381 | -7168 |
| | MMVAE | 36.7 | 32.6 | 32.0 | 21.3 | -1250 | **-1435** | -1372 | -7185 |
| | MoPoE | 33.7 | 29.4 | 31.2 | 20.8 | -1267 | -1505 | -1368 | -7119 |
| | CLAP | **36.7** | **40.4** | **39.9** | **23.4** | **-1242** | -1465 | **-1363** | **-7104** |
| 100 | MVAE | 37.4 | 37.6 | 37.4 | 22.0 | -1295 | -1475 | -1418 | -7327 |
| | MMVAE | 37.1 | 35.0 | 34.0 | 21.7 | -1304 | -1470 | -1418 | -7389 |
| | MoPoE | 35.8 | 36.5 | 36.1 | 21.7 | -1306 | -1499 | -1464 | -7385 |
| | CLAP | **37.6** | **37.7** | **38.4** | **22.3** | **-1282** | **-1466** | **-1406** | **-7313** |

## E.2 Different Number of Missing Modalities.

As the missing modalities are diverse in local clients, we propose to randomly generate the masks on the PolyMNIST dataset. Specifically, we randomly generate different masks with different missing probabilities. We conduct experiments on the PolyMNIST dataset with 3 random masks. From the results in Table 12, our method consistently outperforms other baselines.

Table 12: Imputation coherence accuracy (%), latent space classification and log-likelihood results of some modality subsets on the PolyMNIST dataset.

| | Model | Coherence | | | Latent Classification | | | Likelihood |
|---|---|---|---|---|---|---|---|---|
| | | $M_2$ | $M_4$ | $M_5$ | $\{M_4\}$ | $\{M_4, M_5\}$ | $\{M_2, M_4, M_5\}$ | $X$ |
| mask 1 | MVAE | 18.1 | 69.1 | 18.2 | 80.3 | 74.7 | 77.0 | -6851 |
| | MMVAE | 62.0 | 63.1 | 34.6 | 86.8 | 79.2 | 81.3 | -6951 |
| | MoPoE | 69.5 | 73.1 | 48.3 | 79.0 | 80.5 | 94.7 | -6734 |
| | CLAP | **69.7** | **79.2** | **50.4** | **88.4** | **91.9** | **96.7** | **-6614** |
| mask 2 | MVAE | 22.4 | 70.5 | 16.1 | 82.8 | 71.1 | 77.5 | -6842 |
| | MMVAE | 64.5 | 62.8 | 34.8 | 87.5 | 77.7 | 81.4 | -6951 |
| | MoPoE | 69.8 | 71.9 | 45.2 | 80.3 | 81.7 | 95.6 | -6768 |
| | CLAP | **70.8** | **77.1** | **52.4** | **88.2** | **91.5** | **95.8** | **-6601** |
| mask 3 | MVAE | 15.7 | 69.2 | 15.2 | 81.7 | 74.5 | 77.4 | -6832 |
| | MMVAE | 63.7 | 62.5 | 35.9 | 84.4 | 75.3 | 80.2 | -6960 |
| | MoPoE | 68.4 | 70.7 | 46.4 | 81.3 | 81.4 | 95.0 | -6751 |
| | CLAP | **71.5** | **77.3** | **53.7** | **85.2** | **92.7** | **94.8** | **-6634** |

Besides, we mask the different number of modalities on the eICU dataset. As the results shown in Table 13, the imputation task is more difficult to tackle with more modalities masked.

Table 13: Results on eICU dataset with the different number of modalities masked. MSE and latent classification AUC (%) are averaged over 13 modalities and certain subsets. The log-likelihoods of joint distribution $X$ are reported.

| Masked number | MSE | Latent Classification | Likelihood |
|---|---|---|---|
| 1 | 0.0612 | 65.4 | -321.4 |
| 2 | 0.0653 | 62.3 | -340.2 |
| 3 | 0.0667 | 60.2 | -357.6 |
| 5 | 0.0668 | 54.2 | -382.4 |
| 10 | 0.0705 | 52.1 | -380.7 |

### E.3 Different Latent Vector Dimensionality

We conduct the ablation study using the MNIST-SVHN-TEXT dataset on the latent vector dimensionality. As shown in Table 14, the performance degrades when the latent vector dimensionality is too small (<20). Otherwise, the model performances are similar.

Table 14: Results on MNIST-SVHN-TEXT dataset with the different number of latent vector dimensionality.

| Dimensionality | Coherence | | | Likelihood |
|---|---|---|---|---|
| | $M$ | $S$ | $T$ | $X$ |
| 3 | 45.1 | 19.2 | 78.9 | -1956 |
| 5 | 65.1 | 47.4 | 78.9 | -1919 |
| 10 | 78.0 | 53.4 | 93.1 | -1873 |
| 20 | 95.3 | 58.9 | 92.3 | -1838 |
| 40 | 96.4 | 60.7 | 89.3 | -1847 |
| 60 | 94.4 | 61.1 | 91.7 | -1852 |
| 80 | 95.6 | 60.6 | 91.2 | -1868 |
| 100 | 94.3 | 61.0 | 89.3 | -1876 |

### E.4 Different Level of Statistical Heterogenity

We supplement the ablation study on statistical heterogeneity shown in Table 15. We split the PolyMNIST dataset into 50 clients and each client contains samples from $k$ classes. A smaller $k$ leads to greater heterogeneity. As shown in the Table below, the log-likelihoods decrease with the increase of heterogeneity. The latent classification accuracies are larger for smaller $k$, because smaller $k$ values entail a reduction in the number of classes, making classification easier.

Table 15: Results on the PolyMNIST dataset with different statistical heterogeneity indicator $k$.

| $k$ | Latent Classification | | | | Likelihood | | | |
|---|---|---|---|---|---|---|---|---|
| | $\{M_4\}$ | $\{M_4, M_5\}$ | $\{M_2, M_4, M_5\}$ | Average | $M_2$ | $M_4$ | $M_5$ | $X$ |
| 8 | 31.1 | 33.0 | 31.2 | 19.7 | -1261 | -1483 | -1359 | -7104 |
| 6 | 36.7 | 40.4 | 39.9 | 23.4 | -1242 | -1465 | -1363 | -7104 |
| 4 | 43.5 | 43.9 | 42.9 | 26.7 | -1267 | -1486 | -1384 | -7122 |
| 3 | 50.2 | 52.1 | 52.4 | 30.7 | -1290 | -1487 | -1386 | -7148 |

## F Computation Analysis and Devices

As a generative model, CLAP has a similar number of parameters with other federated multimodal learning methods, including the baselines PoE, MoE, etc. Therefore, CLAP does not bring many

extra computational and communication costs. As we study a novel problem, it is hard to compare the computational overhead with other methods used in other problems. We provide the running-time comparisons with baselines in Table 16, which demonstrates that CLAP does not have too much additional computational overhead with baselines. MoPoE can not work on the eICU dataset due to the vast number ($2^{13}$) of modality subsets. It has an exponential complexity with respect to the number of modalities and costs a vast number of computing resources. Compared with MVAE and MMVAE, our method aggregates the identified several subsets and does not add too much burden on optimization and computation.

**Devices** In the experiments, we conduct all methods on a local Linux server that has two physical CPU chips (Intel(R) Xeon(R) CPU E5-2640 v4 @ 2.40GHz) and 32 logical kernels. All methods are implemented using Pytorch framework and all models are trained on GeForce RTX 2080 Ti GPUs.

Table 16: Run-time consumption comparisons on the eICU dataset

| Methods | Run-time consumption (PolyMNIST) | Run-time consumption (eICU) |
|---|---|---|
| MVAE | 261 min | 70 min |
| MMVAE | 236 min | 61 min |
| MoPoE | 246 min | > 1440 min |
| CLAP | 279 min | 79 min |