# OpenReview forum: "CLAP: Collaborative Adaptation for Patchwork Learning"
_ICLR.cc/2024/Conference — ICLR 2024 spotlight_

### Official Review · Reviewer_QCRo · 2023-10-27

**Soundness:** 4 excellent
**Presentation:** 3 good
**Contribution:** 3 good
**Rating:** 6
**Confidence:** 4

**Summary:**

This paper learns the domain of multimodal learning, with a specific focus on the challenge known as "checkerboard learning." In contrast to conventional federated multimodal learning, where all clients typically share the same modalities, checkerboard learning presents a scenario where different clients may possess different modality combinations. The primary objective of this task is to reconstruct or impute the missing modalities for each client without access to the raw data. To address this problem, the authors propose a method named "Collaborative Adaptation" (CLAP), which encompasses two distinct training stages within each client: Modality Variational Autoencoder (MVAE) and Client-Adaptation Variational Autoencoder (CA-VAE). The authors provide empirical evidence showcasing the efficacy of their proposed approach on various datasets, underscoring its ability to effectively address the challenges posed by checkerboard learning.

**Strengths:**

1. Exploring modality differences has great practical significance in real-world applications. In practice, numerous local clients deal with multi-modal data. Yet, the challenge lies in the substantial inconsistency in the types of modalities. This paper considers a practical and challenging problem.
2. The proposed method is novel and valid for addressing this problem. To impute the incomplete checkerboard, this paper proposes CA-VAE to model the relationships between modalities based on the the assumption that not all modalities are isolated modalities. For the statistical heterogeneity, this paper proposes Pareto Min-Max optimization to balance the performance of all combinations.
3. This paper offers empirical validation of the proposed methodology through a series of experiments. These experiments encompass MNIST datasets, other benckmark datasets and eICU. The results of these experiments underscore the superior performance of the proposed methods when compared to existing approaches.
4. This paper is well-written. The figures is very helpful to understand the motivation of the proposed method.

**Weaknesses:**

1. The proposed method is based on the two assumptions. It seems that these two assumptions are not so strict. Is the effectiveness of the proposed method will be affected by the Assumption 2 (e.g., when there are many missing modalities)?
2. For the challenge of statistical heterogeneity, the author proposes vertical Pareo min-max optimization to address it. Please explain it why the vertical Pareto min-max optimization could mitigate the statistical heterogeneity.
3. In this paper, the author did experiments on some datasets. When the data is more completed, is it still useful to complete the missing modalities?
4. For the statistical heterogeneity, in real-world applications, there are many different types of heterogeneity, how the heterogeneity affects the effectiveness of the proposed method?
5. For the proposed CA-VAE in section 4.4, it seems sound. However, is there direct experimental verification of the effectiveness of CA-VAE?

**Questions:**

Please see the weaknesses above. Besides, I have the following minor problems.

1. In section 4, the author demonstrates the proposed method. However, an algorithm would be more helpful for other to follow this research.
2. For the assumptions in section 3, it is better to give a justification for its practicality.
3. In the conclusion, this paper says that it opens a direction for the future research. Could the author discuss more details about it?

---

> ### Author Response · Authors · 2023-11-17
> **Response to Reviewer QCRo (part 1)**
>
> We would like to thank the reviewer for acknowledging our novelty and the very valuable comments. Below are our responses to the comments in **Weaknesses** and **Questions**.
>
> * **Question 1: The proposed method is based on the two assumptions. It seems that these two assumptions are not so strict. Is the effectiveness of the proposed method will be affected by Assumption 2 (e.g., when there are many missing modalities)?**
>
> **Answer:** Thanks for the insightful comments. We would like to explain it as follows.
>
> **(1). The two assumptions are largely satisfied in reality.** Assumption 1 means that each client has more than one modality. Assumption 2 emphasizes that the missing modalities could be imputed as long as the missing modalities form a connected sub-graph, even if the internal connections are extremely sparse. In real-world scenarios, there often exist clients that possess multimodal data, thereby forming a fully connected subgraph by the involved modalities. The connected subgraph means that the missing modalities in the subgraph of all clients could be imputed.
>
> **(2). The imputation accuracy of CLAP is affected by the number of missing modalities.** In the Sec. E.1 in the Appendix in our original submission, we masked the different number of modalities on the eICU dataset. As the results shown in the following table, when there are more missing modalities, the latent classification accuracy decreases from 65.4 to 52.1, as there are less observable data and the imputation task is more difficult.
>
> | Masked number | MSE   | Latent Classification | Likelihood |
> |---------------|-------|-----------------------|------------|
> | 1             | 0.0612| 65.4                  | -321.4     |
> | 2             | 0.0653| 62.3                  | -340.2     |
> | 3             | 0.0667| 60.2                  | -357.6     |
> | 5             | 0.0668| 54.2                  | -382.4     |
> | 10            | 0.0705| 52.1                  | -380.7     |
>
>
> * **Question 2: For the challenge of statistical heterogeneity, the author proposes vertical Pareo min-max optimization to address it. Please explain why the vertical Pareto min-max optimization could mitigate the statistical heterogeneity.**
>
> **Answer:** Thanks for the valuable advice. We would like to explain it as follows.
>
> (1). **Pareto min-max optimization addresses the statistical heterogeneity compared with simple averaging.** Simple averaging (FedAvg), which is used in the learning scenarios where each client could derive supervised information from their own data (e.g., prediction tasks), rather than entirely obtaining such information from other heterogeneous clients as in checkerboard learning.
>
> Checkerboard learning studies the problem where each client may own incomplete multimodal data even during training. This means that the imputation of modalities for each client is entirely dependent on the dependencies learned from other clients. In this case, a poor dependency learning from one client could hurt the imputation of multiple clients. Because of potential conflicts in multiple dependencies, the learning of particular dependencies could be hurt by simple averaging. Therefore, we propose to use Pareto min-max optimization to ensure that the worst-performing dependency can be learned effectively, so that we could achieve a better imputation performance.
>
> (2). **Experimental verification of the effectiveness of Pareto min-max optimization.** We did ablation studies to verify the effectiveness of the Pareto min-max optimization in our original submission. The detailed experimental results could be found in Sec.5.4 in the maintext.

---

> ### Author Response · Authors · 2023-11-17
> **Response to Reviewer QCRo (part 2)**
>
> * **Question 3: In this paper, the author did experiments on some datasets. When the data is more complex, is it still useful to complete the missing modalities?**
>
> **Answer:** We would like to explain it as follows.
>
> **(1). Compared with baselines, CLAP achieves the best performance on more complicated datasets.** Our experiments are not only conducted on MNIST-type datasets in the original submission. We also provided experiments on CelebA and CUB datasets in the Appendix in our original submission. For example, we show the Imputation coherence and latent space classification results on CelebA dataset in the following table.
>
> | Model  | Coherence (I) | Coherence (T) | Latent Classification (I) | Latent Classification (T) |
> |--------|---------------|---------------|---------------------------|---------------------------|
> | MVAE   | 35.5$_{\pm 0.1}$ | 34.7$_{\pm 0.4}$ | 16.0$_{\pm 0.3}$ | 7.1$_{\pm 0.2}$ |
> | MMVAE  | 34.4$_{\pm 0.2}$ | 32.7$_{\pm 0.3}$ | 43.7$_{\pm 0.2}$ | 41.5$_{\pm 0.1}$ |
> | MoPoE  | 34.9$_{\pm 0.3}$ | 33.7$_{\pm 0.5}$ | 43.8$_{\pm 0.1}$ | 41.2$_{\pm 0.3}$ |
> | CLAP | **37.4**$_{\pm 0.3}$ | **35.2**$_{\pm 0.1}$ | **44.2**$_{\pm 0.1}$ | **45.4**$_{\pm 0.3}$ |
>
>
> CLAP surpasses all baselines and achieves the highest imputation coherence and latent space classification accuracy.
>
> **(2). In more complicated datasets, the performance of all methods decreases.** We copy the imputation coherence of all methods on PolyMNIST dataset from the Sec. 5.1 in the maintext in the following table.
>
> | Model  | $M_1$ | $M_2$ | $M_3$ | $M_4$ | $M_5$ |
> |--------|-------|-------|-------|-------|-------|
> | MVAE   | 22.1$_{\pm 1.1}$ | 19.5$_{\pm 0.5}$ | 15.2$_{\pm 1.4}$ | 68.7$_{\pm 1.2}$ | 19.6$_{\pm 1.0}$ |
> | MMVAE  | 36.3$_{\pm 0.1}$ | 60.5$_{\pm 0.4}$ | 61.3$_{\pm 1.0}$ | 64.4$_{\pm 2.0}$ | 34.4$_{\pm 0.8}$ |
> | MoPoE  | 38.5$_{\pm 0.6}$ | 69.4$_{\pm 1.4}$ | 72.4$_{\pm 1.4}$ | 75.7$_{\pm 0.5}$ | 45.6$_{\pm 1.0}$ |
> | method | **45.2**$_{\pm 0.2}$ | **69.4**$_{\pm 1.1}$ | **74.5**$_{\pm 0.1}$ | **79.3**$_{\pm 0.8}$ | **50.5**$_{\pm 0.3}$ |
>
> The imputation of a more complicated dataset is a more difficult task. Compared with the experimental results on CelebA datasets, all methods on PolyMNIST datasets achieve a higher performance from the two tables above. And CLAP still maintains its superiority compared with other methods.
>
> * **Question 4: For the statistical heterogeneity, in real-world applications, there are many different types of heterogeneity, how the heterogeneity affects the effectiveness of the proposed method?**
>
> **Answer:** Thanks for the very valuable comment. We would like to explain it as follows.
>
> **(1). The statistical heterogeneity studied in our work.** In checkerboard learning, we consider a real-world problem, in which the non-i.i.d. data are typically present and any heterogeneity could potentially exist. For a single modality, local clients can have different marginal distribution, i.e., $P(m^{i}) \not= P(M^{j})$, where $i$ ($j$) denotes the $i$-th ($j$-th) client and $m$ denotes any modality. For multiple modalities, local clients can have different conditional distributions, i.e., $P(m^{i}_{1}| m_{2}) \not= P(m^{j}_{1}| m_{2})$ where $m_{1}$ and $m_{2}$ denote different modalities.
>
> **(2). We did the ablation studies and the likelihood decreases with a higher statistical heterogeneity.** In E.3 in the Appendix in our original submission, we did ablation studies on different levels of statistical heterogeneity. We split the PolyMNIST dataset into 50 clients and each client contains samples from $k$ classes. A small $k$ leads to greater statistical heterogeneity.
>
> | $k$ | Likelihood ($M_2$) | Likelihood ($M_4$) | Likelihood ($M_5$) | Likelihood ($X$) |
> |-----|--------------------|--------------------|--------------------|------------------|
> | 8   | -1261               | -1483               | -1359               | -7104             |
> | 6   | -1242               | -1465               | -1363               | -7104             |
> | 4   | -1267               | -1486               | -1384               | -7122             |
> | 3   | -1290               | -1487               | -1386               | -7148             |
>
> From the table above, the log-likelihoods decrease with the increase of heterogeneity.

---

> ### Author Response · Authors · 2023-11-17
> **Response to Reviewer QCRo (part 3)**
>
> * **Question 5: For the proposed CA-VAE in section 4.4, it seems sound. However, is there direct experimental verification of the effectiveness of CA-VAE?**
>
> **Answer:** Thanks for the kind advice about the experimental verification of CA-VAE. We would like to clarify that CA-VAE learns the dependencies between all common observable modalities and the missing modalities. For each missing modality imputation, the CA-VAE identifies the dependency which maximizes the utilization of the observable modalities. We also conduct an ablation study on eICU in the following to demonstrate it.
>
> | Model         | MSE    | Latent Classification | Likelihood |
> |---------------|--------|------------------------|------------|
> | CLAP          | 0.0653 | 62.3                   | -340.2     |
> | w/o rule(7)   | 0.0684 | 58.4                   | -348.5     |
>
>
> From the above table, CA-VAE significantly improves the performance. Therefore, in most cases, CLAP could achieve better imputation quality.
>
> * **Question 6: In section 4, the author demonstrates the proposed method. However, an algorithm would be more helpful for others to follow this research.**
>
> **Answer:** We would like to thank the reviewer for the very helpful advice. As you mentioned, an algorithm would be more helpful for understanding our framework. We did present our algorithm in Sec. C.1 in Appendix in our original submission. Because of the page limit, we put it in the Appendix. To ensure clarity for other reviewers, we have incorporated your suggestions into the final version of our paper, placing the algorithm in the main body of the text.
>
> * **Question 7: For the assumptions in section 3, it is better to give a justification for its practicality.**
>
> **Answer:** Thank the reviewer for the professional advice. For the justification of the assumptions in Sec. 3, we would like to explain it as follows. In real-life scenarios, data is often generated and stored in a multimodal fashion. However, it is challenging for individual clients to possess data from all modalities simultaneously. Therefore, the effective utilization of data dispersed across different clients, where the modalities may vary, poses a crucial and practical challenge.
>
> **(1). the practicality of Assumption 1.**
> Assumption 1 states that each client must have data from at least one modality, implying that the dataset for a given client should not be empty. This assumption is grounded in reality, as clients participating in collaborative modeling must possess data of at least one modality.
>
> **(2). the practicality of Assumption 2.** Assumption 2 states that the modalities within the graph formed by each client should constitute a connected subgraph. This assumption, like the previous one, aligns with reality because it implies that all modalities cannot be entirely isolated. Therefore, as long as one client includes two or more modalities, a connected subgraph exists.
>
> * **Question 8: In the conclusion, this paper says that it opens a direction for future research. Could the author discuss more details about it?**
>
> **Answer:** We aim to present our perspectives on this subject and encourage scholarly discourse. Our research centers on the effective utilization of distributed multimodal data characterized by inconsistent modality combinations. This contribution offers valuable insights into the realm of multi-modal learning.
>
> In contrast to prevalent multimodal learning approaches that strive to align representations across multiple modalities, often necessitating a complete checkerboard structure and abundant data, we address the practical challenge posed by sparse and costly multimodal datasets. Our proposed method, CLAP, introduces an approach enabling model learning from an incomplete multimodal checkerboard. This methodology maximizes data utilization during the training of expansive multimodal models. The implications of our work extend to the broader landscape of advancing large multi-modal model learning.

---

### Official Review · Reviewer_ccS1 · 2023-10-29

**Soundness:** 3 good
**Presentation:** 3 good
**Contribution:** 4 excellent
**Rating:** 8
**Confidence:** 4

**Summary:**

The authors explore a novel federated multi-modal learning scenario “checkerboard learning”. Different from existing research which mostly assume the presence of complete modalities, the proposed task considers a practical problem where different clients possess various data modalities. The federated clients aim to impute the missing modalities by collaborating with other clients. A client could impute the missing modalities relying on the dependency learned in other clients which contain the aiming modalities. The proposed framework CLAP to tackle this problem consists of modality VAEs and client-adaptation VAEs. The modality VAEs strive to address the statistical heterogeneity problem by balancing the distribution distance among clients. And the client-adaptation VAEs are the additional decoders used for balancing the modality heterogeneity among clients. Experimental results demonstrate the superiority of CLAP.

**Strengths:**

1. The authors propose a very novel task. Existing federated multi-modal learning research mostly assume the same modality combination in all clients. The scenario of different modalities among clients is under-explored. Besides, the proposed task is substantially practical, as collaborative scenarios involving multiple clients with diverse modality combinations are frequently encountered in real-world applications.
2. In the proposed new task, the missing modalities are not only invisible during testing, but also unavailable during training, making the checkerboard learning a much more challenging and practical problem.
3. The proposed method is concise and effective. The method is developed based on a comprehensive study of the checkerboard learning problem, wherein the challenges can be concluded into two aspects: statistical heterogeneity and modality combination heterogeneity among clients. The authors employ the Pareto min-max framework to address the aforementioned heterogeneity, which proves to be fundamentally valid.
4. They conduct extensive experiments with sufficient datasets and compare with various baselines. They also perform experiments on a real-world clinical dataset, which demonstrates the effectiveness of the method in practice.

**Weaknesses:**

1. This paper studies a useful and significant learning task. From my knowledge, it is the first to investigate checkerboard learning. I want to know whether there is previous related research.
2. There are modality VAEs and client-adaptation VAEs in the proposed framework. These VAEs are specifically designed to tackle two different challenges: statistical heterogeneity and modality heterogeneity, respectively. One important question is whether they share the same encoders. If they do, it is crucial to understand how they interact and influence each other.
3. The details behind certain aspects of the proposed method needs more explanation. More clarification is needed regarding why the employment of the Pareto min-max framework is advantageous for addressing heterogeneity, as well as the detailed implementation of this framework.
4. During the application, the proposed method incorporates an additional set of decoders. Does it entail a significant increase in computational overhead?

**Questions:**

1. This paper studies a useful and significant learning task. From my knowledge, it is the first to investigate checkerboard learning. I want to know whether there are previous related research?
2. There are modality VAEs and client-adaptation VAEs in the proposed framework. These VAEs are specifically designed to tackle two different challenges: statistical heterogeneity and modality heterogeneity, respectively. One important question is whether they share the same encoders. If they do, it is crucial to understand how they interact and influence each other.
3. The details behind certain aspects of the proposed method needs more explanation. More clarification is needed regarding why the employment of the Pareto min-max framework is advantageous for addressing heterogeneity, as well as the detailed implementation of this framework.
4. During the application, the proposed method incorporates an additional set of decoders. Does it entail a significant increase in computational overhead?

---

> ### Author Response · Authors · 2023-11-17
> **Response to Reviewer ccS1 (part 1)**
>
> We would like to thank the reviewer for the positive and professional comments. Below are our responses to the comments in **Weaknesses** and **Questions**.
>
> * **Question 1: This paper studies a useful and significant learning task. From my knowledge, it is the first to investigate checkerboard learning. I want to know whether there are previous related research?**
>
> **Answer:** For the previous related research, we would like to explain it as follows.
>
> **(1). there are previous research studying the imputation of the missing modalities in a traditional learning scenario.** Existing research (e.g., [1,2,3,]) propose to yield generalizable representations to capture the relationship across modalities, where a subgroup of examples own all modalities. In these work, multimodal representation learning methods were proposed for traditional learning scenario, as there are no distribution discrepancies across different clients.
>
> **(2). There are a few research investigating the task of distributed multimodal learning with all modalities observable.** Some researchers propose to collaboratively learn models from multimodal data [4,5]. During training, all clients own the data with all modalities.
>
> **(3). Checkerboard learning cares about the learning scenarios where local clients only have incomplete modalities.** In our paper, we are interested in addressing a practical and upstream learning task: how to impute a checkerboard when local clients have various incomplete multimodal data. Compared with existing modality imputation work, checkerboard learning is more challenging because of the distribution shift. Meanwhile, Checkerboard learning learns models from incomplete multimodal data during training, while existing distributed multimodal learning learns models from the complete multimodal data.
>
> [1]. Mike Wu and Noah Goodman. Multimodal generative models for scalable weakly-supervised
> learning. Advances in Neural Information Processing Systems, 31, 2018.
>
> [2]. Yuge Shi, Brooks Paige, Philip Torr, et al. Variational mixture-of-experts autoencoders for multimodal deep generative models. Advances in Neural Information Processing Systems, 32, 2019.
>
> [3]. Thomas M Sutter, Imant Daunhawer, and Julia E Vogt. Generalized multimodal elbo. In International Conference on Learning Representations, 2020b.
>
> [4]. Yuchen Zhao, Payam Barnaghi, and Hamed Haddadi. Multimodal federated learning. arXiv preprint arXiv:2109.04833, 2021.
>
> [5]. Yu Q, Liu Y, Wang Y, et al. Multimodal Federated Learning via Contrastive Representation Ensemble[C]//The Eleventh International Conference on Learning Representations. 2022.
>
> * **Question 2: There are modality VAEs and client-adaptation VAEs in the proposed framework. These VAEs are specifically designed to tackle two different challenges: statistical heterogeneity and modality heterogeneity, respectively. One important question is whether they share the same encoders. If they do, it is crucial to understand how they interact and influence each other.**
>
> **Answer:** We would like to explain it as follows.
>
> (1). **Modality VAEs and CA-VAEs share the same encoders, and have different decoders to mitigate the potential conflict.** As shown in Figure 3 in the maintext, modality VAE and CA-VAE share the same encoder ***E***. Modality VAE has the decoder $D^{M}$ and CA-VAE has the decoder $D^{C}$.
>
> **Modality VAE.** We propose to generate unbiased representations with modality VAEs by Pareto min-max optimization. In this way, we minimize the maximal KL divergence on the worst client, leading to more uniform KL divergence across all clients.
>
> **CA-VAEs.** With the generated unbiased representations from the modality VAE, CA-VAE proposes to model the modality dependencies between the intersection and the remaining modalities, which maximizes the utilization of the observable information.
>
> (2). **How they interact and influence each other:** modality VAE translates multi-modal data into fixed dimensional representations, which are used for the imputation by CA-VAE. In the implementation of CALP, we train modality VAE firstly to generate the representations with a fixed dimension for all modalities. Then we fix the encoder $E$ and train CA-VAE to optimize the decoder $D^{C}$ to complete the checkerboard.

---

> > ### Comment · Reviewer_ccS1 · 2023-11-22
> > **Thank you**
> >
> > I greatly appreciate the authors' dedicated explanation, all of my concerns have been well addressed and I will keep my original score.

---

> ### Author Response · Authors · 2023-11-17
> **Response to Reviewer ccS1 (part 2)**
>
> * **Question 3: The details behind certain aspects of the proposed method need more explanation. More clarification is needed regarding why the employment of the Pareto min-max framework is advantageous for addressing heterogeneity, as well as the detailed implementation of this framework.**
>
> **Answer:** We would like to explain it as follows.
>
> **(1) Pareto min-max optimization is advantageous for addressing heterogeneity.** Checkerboard learning studies the problem where each client may own incomplete multimodal data even during training. This means that the imputation of modalities for each client is entirely dependent on the dependencies learned from other clients. In this case, a poor dependency learning from one client could hurt the imputation of multiple clients. Because of potential conflicts in multiple dependencies, the learning of particular dependencies could be hurt by simple averaging. Therefore, we propose to use Pareto min-max optimization to ensure that the worst-performing dependency can be learned effectively, so that we could achieve a better imputation performance. Experiments in Sec. 5.4 verify the effectiveness of Pareto min-max optimization for the imputation of the checkerboard.
>
> **(2). the detailed implementation is shown in the Algorithm 1 in the Appendix.** We are sorry that we put the algorithm of CLAP in the Appendix in our original submission because of the page limit, making it appear somewhat complex. Compared with baselines (e.g., MVAE[1], MoPoE[2]), our framework can also be effortlessly implemented in two steps. Specifically, (1). train the Modality VAE with vertical Pareto min-max optimization;
> (2). fix the encoder of the Modality VAE and train CA-VAE with horizontal Pareto min-max optimization.
>
>
> * **Question 4: During the application, the proposed method incorporates an additional set of decoders. Does it entail a significant increase in computational overhead?**
>
> **Answer:** We would like to explain that our framework CLAP has no significant extra computational overhead compared with other related methods.
>
> **(1). the additional set of decoders in the framework CLAP is used for reconstructing the input modality, which does not bring too much computational overhead.** As a generative model, CLAP has a similar number of parameters with other federated multimodal learning methods, including the baselines PoE, MoE, etc. Therefore, CLAP does not bring many extra computational and communication costs. Meanwhile, in the realization of CLAP, the decoders in modality VAE reconstruct the input modality data, which has a relatively small computational overhead.
>
>
> **(2). the run-time consumption comparisons in our original submission show that CLAP has a similar computational overhead compared with baselines.** In the Sec. F in the appendix in our original submission, we provided the run-time consumption comparisons. We copy the results in the following for your easy reference.
>
> | Methods | Run-time consumption (PolyMNIST) | Run-time consumption (eICU) |
> | :--: | :--: | :--: |
> | MVAE | 261 min | 70 min        |
> | MMVAE | 236 min | 61 min       |
> | MoPoE | 246 min |  > 1440 min       |
> | CLAP | 279 min |  79 min       |
>
> From the table above, MoPoE can not work on the eICU dataset due to the vast number ($2^{13}$) of modality subsets. It has an exponential complexity with respect to the number of modalities and costs a vast number of computing resources. Compared with MVAE and MMVAE, our method aggregates the identified several subsets and does not add too much burden on optimization and computation.

---

### Official Review · Reviewer_scKN · 2023-10-30

**Soundness:** 3 good
**Presentation:** 3 good
**Contribution:** 4 excellent
**Rating:** 8
**Confidence:** 5

**Summary:**

The authors introduce a novel problem in machine learning referred to as the "Checkerboard Learning" This problem focuses on the issue of missing modalities within local client datasets by employing multimodal learning imputation techniques. To tackle this problem, the authors present a comprehensive framework called "Collaborative Adaptation" or CLAP. In essence, CLAP aims to bridge the gap between differing data distributions and unearth interdependencies between local clients through the utilization of two VAEs. A pivotal component of CLAP is the Modality VAE, designed to enhance the consistency of data representation by implementing a shared Modality Encoder across all clients. The Client-Adaptation VAE (CA-VAE) builds upon the Modality VAE by introducing a new decoder to learn the dependencies among different modalities. Empirical experiments demonstrate that CLAP exhibits strong performance across both benchmark and real-world datasets, validating its effectiveness in addressing the checkerboard learning challenge.

**Strengths:**

1. The proposed method is very practical. In real-world, there are many local clients with multi-modal data. However, the modality types are largely inconsistent, makes it difficult to learn models directly. This paper proposes “checkerboard learning”, which formulates this problem precisely.

2. The analysis of the problem is important, and the proposed method is convincing. This paper analyzes the checkerboard learning deeply and summarizes three challenges of checkerboard learning, e.g., heterogeneity, different modality combinations. To complete the checkerboard, this paper proposes two basic assumptions, which makes it possible to impute the missing modalities by learning from other clients.

3. This paper provides experimental verification of the proposed method. It conducts experiments on MNIST-typed datasets, CelebA CUB and a real-world datasets eICU. The experiments show the superiority of the proposed methods compared with existing methods.

4. In the Appendix, the authors did sufficient ablation studies for the proposed methods, including more clients and the number of missing modalities. It is very helpful to evaluate the proposed method.

**Weaknesses:**

1.In related work, most of the papers are published before 2022. Are there more recent related works which study the similar learning scenarios?

2.This paper experimentally verifies the effectiveness of the proposed Pareto min-max optimization. Could the authors give more explanation of it? Why the Pareto min-max optimization is helpful for the imputation of the checkerboard?

3.For the real-world application, the implementation is very important. From the C.3 in Appendix, the author discusses the limitations. How to address the privacy issue when in real-world applications?

**Questions:**

1. In related work, most of the papers are published before 2022. Are there more recent related works which study the similar learning scenarios?

2. This paper experimentally verifies the effectiveness of the proposed Pareto min-max optimization. Could the authors give more explanation of it? Why the Pareto min-max optimization is helpful for the imputation of the checkerboard?

3. The algorithm is in Appendix. I guess it is because of the page limit. For the real-world application, the implementation is very important. From the C.3 in Appendix, the author discusses the limitations. How to address the privacy issue when in real-world applications?

---

> ### Author Response · Authors · 2023-11-17
> **Response to Reviewer scKN**
>
> We would like to thank the reviewer for recognizing our contributions. Below are our responses to the comments in **Weaknesses** and **Questions**.
>
> * **Question 1: In related work, most of the papers are published before 2022. Are there more recent related works that study the similar learning scenarios?**
>
>
> **Answer:** We would like to explain that **there are no existing related methods directly available for our proposed problem**.
>
> **(1). Existing multimodal imputation learning requires all modalities to be observable during training[1].** Existing related multimodal learning methods focus on learning a representation to impute the missing modalities during inference. It requires all modalities to be observable during training. However, in checkerboard learning, there is no client who owns the data of all modalities even during training.
>
> **(2). Existing federated multimodal learning requires clients to have data for all modalities[2,3].** There are some recent works in federated multimodal learning, where the task is how to use multimodal data for prediction. These works also require clients to possess data across all modalities during training, which cannot be directly used for the imputation when all clients have incomplete multimodal data in checkerboard learning.
>
> [1]. Mike Wu and Noah Goodman. Multimodal generative models for scalable weakly-supervised learning. Advances in Neural Information Processing Systems, 31, 2018.
>
> [2]. Yuchen Zhao, Payam Barnaghi, and Hamed Haddadi. Multimodal federated learning. arXiv preprint arXiv:2109.04833, 2021.
>
> [3]. Yu Q, Liu Y, Wang Y, et al. Multimodal Federated Learning via Contrastive Representation Ensemble[C]//The Eleventh International Conference on Learning Representations. 2022.
>
> * **Question 2: This paper experimentally verifies the effectiveness of the proposed Pareto min-max optimization. Could the authors give more explanation of it? Why the Pareto min-max optimization is helpful for the imputation of the checkerboard?**
>
> **Answer:** We would like to explain the effectiveness of the Pareto min-max optimization compared to simple averaging as follows.
>
> **(1). Simple averaging.** Averaging (e.g., in FedAvg) is used in the learning scenarios where each client could derive supervised information from their own data (e.g., prediction tasks), rather than entirely obtaining such information from other heterogeneous clients as in checkerboard learning.
>
> **(2). Pareto min-max optimization.** Checkerboard learning studies the problem where each client may own incomplete multimodal data even during training. This means that the imputation of modalities for each client is entirely dependent on the dependencies learned from other clients. In this case, a poor dependency learning from one client could hurt the imputation of multiple clients. Because of potential conflicts in multiple dependencies, the learning of particular dependencies could be hurt by simple averaging. Therefore, we propose to use Pareto min-max optimization to ensure that the worst-performing dependency can be learned effectively, so that we could achieve a better imputation performance.
>
> **(3). Experimental verification.** We did ablation studies to verify the effectiveness of the Pareto min-max optimization in our original submission. The detailed experimental results can be found in Sec.5.4 in the maintext.
>
> * **Question 3: For the real-world application, the implementation is very important. From the C.3 in Appendix, the author discusses the limitations. How to address the privacy issue when in real-world applications?**
>
> **Answer:** Thanks for the professional comments. We would like to explain that CLAP has no extra privacy risks compared with other federated generative methods. Specifically,
>
> (1). **client-to-client: there is no information exchange between clients.** In the implementation of CLAP, each client computes and optimizes its objective using its data. There is no information transmission across clients directly.
>
> (2). **client-to-server: only the sum of the gradients is allowed to be sent to the server.** The learned representations are protected by local clients throughout the training process. Only the sum of the gradient information is allowed to be transmitted to the server. Then the server sends the averaged gradient to local clients, which avoids the information exchange across clients directly. Therefore, compared with existing federated generative methods (e.g., [2]), there are no extra privacy risks in CLAP.
>
> (3). **More techniques on privacy preservation may need further exploration in the federated generation problem.** To further protect privacy during generation, existing techniques may be applied in federated generative models, for example, block-chain, differential privacy, secure multi-party computation, etc.

---

### Official Review · Reviewer_DjdF · 2023-11-06

**Soundness:** 3 good
**Presentation:** 3 good
**Contribution:** 3 good
**Rating:** 8
**Confidence:** 2

**Summary:**

This paper primarily addresses the task of processing multimodal data in a multi-client context, proposing the CLAP framework. The framework consists of a modality VAE and a client-adaptation VAE. The modality VAE learns a representation for each modality, while the client-adaptation VAE handles dependencies between modalities. Given that data are from multiple clients and not independently and identically distributed, to alleviate the statistical heterogeneity of the learned representations, an encoder is shared and the KL divergence of the representations is balanced by maximizing the maximum KL divergence. The authors demonstrate through experiments that the CLAP framework effectively processes heterogeneous data while protecting user privacy.

**Strengths:**

1. The CLAP framework proposed in this article achieves the imputation of multimodal data when dealing with non-independent and identically distributed data.
2. By employing a vertical Pareto min-max optimization strategy to balance the KL divergence, CLAP addresses the issue of statistical heterogeneity in client data distributions.
3. The training of CLAP adheres to the federated learning framework, effectively protecting the privacy of users across multiple clients.

**Weaknesses:**

1. The framework presented in this paper is complex; is it suitable for practical deployment?
2. The experimental section includes a limited number of medical datasets, which constrains the assessment of the model.

**Questions:**

1. Why is client data non-independent and identically distributed? How is the feature distribution of the same modality dataset across different clients demonstrated?
2. The author presents a heterogeneous data exchange among multiple medical institutions, but I am not quite clear whether the complexity of the current research algorithm is already suited to the needs of the current research context.

---

> ### Author Response · Authors · 2023-11-17
> **Response to Reviewer DjdF (part 1)**
>
> We would like to thank the reviewer for the very insightful and valuable comments. Below are our responses to the comments in **Weaknesses** and **Questions**.
>
> * **Question 1: The framework presented in this paper is complex; is it suitable for practical deployment?**
>
> **Answer:** Thank the reviewer for the valuable comments. We would like to explain it as follows.
>
> **(1). The CLAP framework can be easily implemented.** We are sorry that we put the algorithm of CLAP in the Appendix in our original submission because of the page limit, making it appear somewhat complex. Compared with baselines (e.g., MVAE[1], MoPoE[2]), our framework can also be effortlessly implemented in two steps. Specifically, (1). train the Modality VAE with vertical Pareto min-max optimization;
> (2). fix the encoder of the Modality VAE and train CA-VAE with horizontal Pareto min-max optimization. Compared with baselines, our algorithm has a similar computational overhead as shown in the following table.
>
> | Methods | Run-time consumption (PolyMNIST) | Run-time consumption (eICU) |
> | :--: | :--: | :--: |
> | MVAE | 261 min | 70 min        |
> | MMVAE| 236 min | 61 min        |
> | MoPoE | 246 min |  > 1440 min  |
> | CLAP | 279 min |  79 min       |
>
> From the table above, our method introduces a negligible burden on optimization and computation.
>
> [1]. Mike Wu and Noah Goodman. Multimodal generative models for scalable weakly-supervised
> learning. Advances in Neural Information Processing Systems, 31, 2018.
> [2]. Yuge Shi, Brooks Paige, Philip Torr, et al. Variational mixture-of-experts autoencoders for multimodal deep generative models. Advances in Neural Information Processing Systems, 32, 2019.
>
> **(2). CLAP is inherently designed for practical deployment.** The primary motivation behind checkerboard learning is to address real-world requirements. **Motivation:** In real-world, the multi-modal data are typically sparse and expensive. A key problem is how to use the multi-modal data with different modality combinations. Checkerboard learning proposes to impute the incomplete checkerboard for various downstream tasks. **Privacy:** Data privacy has become an escalating concern. In the realization of CLAP, we collaboratively learn models without directly accessing the raw data from other clients. **Efficiency:** CLAP has a similar number of parameters with other federated multimodal while maintaining the best performance. We experimentally show that CLAP has a similar computation overhead compared with others.
>
> * **Question 2: The experimental section includes a limited number of medical datasets, which constrains the assessment of the model.**
>
> **Answer:** We would like to thank the reviewer for the insightful comments. We would like to explain it as follows.
>
> **(1). we follow the setting in baselines to assess our method on all possible benchmarks.** In our work, we assess our method following the setting in baselines [1,2,3] and did experiments on all benchmarks, including PolyMNIST, MNIST-SVHN-TEXT, CelebA, and CUB datasets. Experiments on the benchmarks verify the effectiveness of our proposed method.
>
> [1]. Mike Wu and Noah Goodman. Multimodal generative models for scalable weakly-supervised
> learning. Advances in Neural Information Processing Systems, 31, 2018.
> [2]. Yuge Shi, Brooks Paige, Philip Torr, et al. Variational mixture-of-experts autoencoders for multimodal deep generative models. Advances in Neural Information Processing Systems, 32, 2019.
> [3]. Thomas M Sutter, Imant Daunhawer, and Julia E Vogt. Generalized multimodal elbo. In International Conference on Learning Representations, 2020b.
>
> **(2). Besides of the experiments on a medical dataset in the maintext, we also conduct other experiments on eICU in the Appendix.** Because of privacy concerns, it is hard to have access to a real-world clinical dataset gathered from multiple hospitals. Compared with the baselines, we are the first to assess the method on a real-world public medical dataset eICU. Besides, we also did experiments on eICU by dividing the rest 12 modalities into 4 modality groups, and the results are shown in the Sec. D.2 in the Appendix in our original submission.

---

> ### Author Response · Authors · 2023-11-17
> **Response to Reviewer DjdF (part 2)**
>
> * **Question 3: Why is client data non-independent and identically distributed? How is the feature distribution of the same modality dataset across different clients demonstrated?**
>
> **Answer:** Thanks for the detailed comments. We would like to explain it as follows.
>
> **(1). Statical heterogeneity (i.e. non-independent and identically distributed data) has been a key challenge in distributed/federated learning[1,2] and has garnered widespread attention from researchers.** Non-i.i.d data typically emerge in federated learning scenarios as data could be gathered from a heterogeneous group of users in reality. For example, hospitals in different areas have various demographic populations, which causes a severe distribution shift[3].
>
> **(2). For the different feature distributions across different clients, we set the feature distribution following the setting in baselines.** For the feature distribution, we set the non-i.i.d environment following the baselines[4]. For example, for MNIST datasets, local clients are assigned different classes of data. For example, the first client is assigned the data with the first 5 labels. The second client is assigned the data with the second 5 labels.
>
> For the experiments on the real-world dataset eICU, the data is collected from different hospitals. The hospital collaboration network is a typical checkerboard learning scenario. Therefore, we treat each hospital as a natural client.
>
> [1]. Brendan McMahan, Eider Moore, Daniel Ramage, Seth Hampson, and Blaise Aguera y Arcas.
> Communication-efficient learning of deep networks from decentralized data. In Artificial intelligence and statistics, pp. 1273–1282. PMLR, 2017.
> [2]. Li T, Sahu A K, Talwalkar A, et al. Federated learning: Challenges, methods, and future directions[J]. IEEE signal processing magazine, 2020, 37(3): 50-60.
> [3]. Rajendran S, Xu Z, Pan W, et al. Data heterogeneity in federated learning with Electronic Health Records: Case studies of risk prediction for acute kidney injury and sepsis diseases in critical care[J]. PLOS Digital Health, 2023, 2(3): e0000117.
> [4]. McMahan B, Moore E, Ramage D, et al. Communication-efficient learning of deep networks from decentralized data[C]//Artificial intelligence and statistics. PMLR, 2017: 1273-1282.
>
> * **Question 4: The author presents a heterogeneous data exchange among multiple medical institutions, but I am not quite clear whether the complexity of the current research algorithm is already suited to the needs of the current research context.**
>
> **Answer:** We would like to thank the reviewer for the insightful comments about the practicability of our work. We guess this question is the same as **Question 1.** Please refer to the **Answer** in **Question 1.** If we misunderstand this problem, please feel free to point it out at any time.

---

> > ### Comment · Reviewer_DjdF · 2023-11-17
> >
> > Thank you very much for the author's response. The responses (part 1 and part 2) have resolved my doubts. Combined with the suggestions of other reviewers and the author's responses to them, I believe that this article has provided very novel methods technically, and also has a very promising prospect in clinical applications. Therefore, I have revised my rating of this article.

---

### Meta-Review · Area_Chair_vm1B · 2023-12-05

**Metareview:**

This paper proposes a distributed learning setting with multimodal data at each client where each client has access to data only from a subset of the modalities.

The paper received positive reviews from all the reviewers who appreciated the paper for the problem setting it considered, the proposed method, and its evaluation.

Some of the concerns raised by the reviewers were also adequately addressed by the authors' rebuttal.

Based on the reviews and my own reading of the paper, I recommend acceptance.

**Justification For Why Not Higher Score:**

Accept with spotlight is adequate given its interesting problem setting, but the novelty/appeal is not as high so merit an oral.

**Justification For Why Not Lower Score:**

The problem setting is interesting and the paper should receive at least a spotlight presentation.

---

### Decision · Program_Chairs · 2024-01-16

Accept (spotlight)